# High-dimensional phenotyping to define the genetic basis of cellular morphology

Matthew Tegtmeyer [1,2,3,17], Jatin Arora[4,5,6,7,17], Samira Asgari [4,5,6,7,8,9,17], Beth A. Cimini [10], Ajay Nadig[1,7,11], Emily Peirent[1], Dhara Liyanage[1], Gregory P. Way [10], Erin Weisbart [10], Aparna Nathan [4,5,6,7,8], Tiffany Amariuta[8,12,13,14], Kevin Eggan [1,2], Marzieh Haghighi [10], Steven A. McCarroll [1,8,15], Luke O'Connor [7,8], Anne E. Carpenter [10], Shantanu Singh [10] ✉, Ralda Nehme [1,2] ✉ & Soumya Raychaudhuri [4,5,6,7,8,16] ✉

The morphology of cells is dynamic and mediated by genetic and environmental factors. Characterizing how genetic variation impacts cell morphology can provide an important link between disease association and cellular function. Here, we combine genomic sequencing and high-content imaging approaches on iPSCs from 297 unique donors to investigate the relationship between genetic variants and cellular morphology to map what we term cell morphological quantitative trait loci (cmQTLs). We identify novel associations between rare protein altering variants in *WASF2*, *TSPAN15*, and *PRLR* with several morphological traits related to cell shape, nucleic granularity, and mitochondrial distribution. Knockdown of these genes by CRISPRi confirms their role in cell morphology. Analysis of common variants yields one significant association and nominate over 300 variants with suggestive evidence (P < 10⁻⁶) of association with one or more morphology traits. We then use these data to make predictions about sample size requirements for increasing discovery in cellular genetic studies. We conclude that, similar to molecular phenotypes, morphological profiling can yield insight about the function of genes and variants.

Cellular morphology is an important and informative cellular trait in a variety of biological contexts, especially the study of disease. A classic example is sickle cell anemia, which is named for the sickle-like morphology of blood cells observed in patients afflicted with this condition[1]. Like other traits such as gene expression, cellular morphology is mediated by genetic variation. Genetic studies have implicated various loci associated with red blood cell phenotypes such as mean volume and hemoglobin content[2,3]. However, there is still limited understanding of how human genetic diversity shapes cell morphology. Profiling cell morphology in different cell types and across genetically diverse populations could facilitate the identification of morphology-associated genetic variants.

Induced pluripotent stem cells (iPSCs) provide a powerful tool for capturing genetic diversity in living biological systems and large publicly or commercially available collections provide access to cell lines from donors of diverse ancestry and genetic backgrounds[4–9]. These collections have enabled the study of how human common and rare genetic variation impacts cellular function and behavior, with a focus on gene expression and chromatin accessibility phenotypes[10–15]. Studies exploring genetic factors that drive cell morphology have shown promise but are limited by sample size and the resolution by which morphological traits are quantified[16]. Additional efforts with increased sample sizes and greater resolution of cell morphology

---

measurements are critical to expanding discovery power for genetic studies of cellular phenotypes.

Innovations in microscopy and image analysis have enabled the measurement of thousands of morphological traits from a single cell, constructing morphology based 'profiles'. Cell Painting, for example, leverages multiplexed dyes to enable the measurement of traits across many cellular compartments and organelles[17,18]. Cell Painting can ascertain gene function by linking expression to cellular traits and has been used to enable the prediction of functional impacts from lung cancer variants[19,20]. Cell morphology profiling provides a great asset for functional genomics studies compared to methods such as gene expression, being much more affordable and easily scalable at the bulk and single cell level. We hypothesized this approach could be leveraged in combination with iPSC technology to elucidate relationships more broadly between cell morphology and genetic variants.

Here, we identified the morphological impacts of genomic variants, or cell morphological quantitative trait loci (cmQTLs), by generating high-throughput morphological profiling and whole genome sequencing data on iPSCs from 297 unique donors. Leveraging Cell Painting data on >5 million iPSCs derived from these donors, we quantified 3418 cell morphological traits and assessed their associations with rare and common genetic variants genome-wide. We identified trait-associations with rare-variant burden in several genes including *WASF2*, *PRLR*, and *TSPAN15* which we then functionally validate using CRISPR interference. Additionally, we nominated one common variant convincingly associated with morphology and found suggestive evidence for over 300 loci. Finally, we leveraged these results to make predictions about sample size requirements for increasing discovery power for both common and rare variants in future cellular genetic studies. These findings show that similar to gene expression, the morphology of cells is mediated by genetic variation and highlights the utility of image-based methods for functional genomics.

## Results

### Morphological profiling and whole-genome sequencing on iPSCs from 297 unique donors

To study associations between genetic variants and morphological traits, we assembled a cohort of iPSC lines from 297 unique donors for which we generated image-based profiling and whole-genome sequencing data (Fig. 1). We obtained pre-derived cell lines from the CIRM iPSC repository[5]. Age, sex, medical history, ethnicity, and

relatedness to other samples were recorded using questionnaires at time of enrolment and sample collection. Each iPSC line was subjected to a pluripotency test as well as genotyping to identify any abnormal karyotypes. Upon receipt, we expanded and cryo-banked each iPSC line, and performed genotyping (using the Global Screening Array (GSA)) and 30X whole-genome sequencing (WGS) on all lines. Any cell lines displaying abnormal karyotypes or genomic rearrangements >1 Mb were excluded from our study. The final cohort used in this work included 297 distinct donors of which 153 were male and 144 were female, with an average age of 21 ± 10 (sd) years. Of the 297 donors, 207 had self-reported ancestry of European and 90 individuals reported non-European (Table 1). IPSC lines were generated from B-cells or fibroblasts using a non-integrating episomal vector system previously described[5] (Table 1). All donors included in this study have been properly consented for iPSC derivation, the experiments performed in this work, and genomic data sharing. We performed a principal component analysis (PCA) to observe the genetic diversity of cells utilized in our collection (Fig. S1A). A summary breakdown of our cohort is included in Table 1 and individual cell line level metadata is included Table S1.

To quantify cellular traits, we adopted the Cell Painting assay[17,18]. This multiplexing dye assay uses six stains to capture morphological characteristics for eight cellular compartments: Hoechst 33342 (DNA), wheat germ agglutinin (WGA) (golgi and plasma membrane), concanavalin A (endoplasmic reticulum), MitoTracker (mitochondria), SYTO 14 (nucleoli and cytoplasmic RNA), and phalloidin (actin). Images are processed using the open-source CellProfiler software to extract thousands of features of each cell's morphology such as shape, intensity, and texture statistics, thus forming a high-dimensional profile for each single cell[21].

We generated Cell Painting data from all 297 donors leveraging a systematic workflow to ensure cells were treated in identical fashion across all rounds of imaging. Cell lines were thawed in batches of 48 and passaged 3 days later into a 96-well deep well plate before being transferred into a 384-well screening plate using an automated liquid handling device (Fig. S1B, Methods). Cells were plated at a density of 10k cells/per well and fixed 6 h post-plating, so as to allow for cell attachment while minimizing differences in cell growth rates, which we observed during cell line expansion (Fig. S1C). We determined these conditions through a pilot screen that contained 6 cell lines plated across various densities and fixation timepoints, which showed we

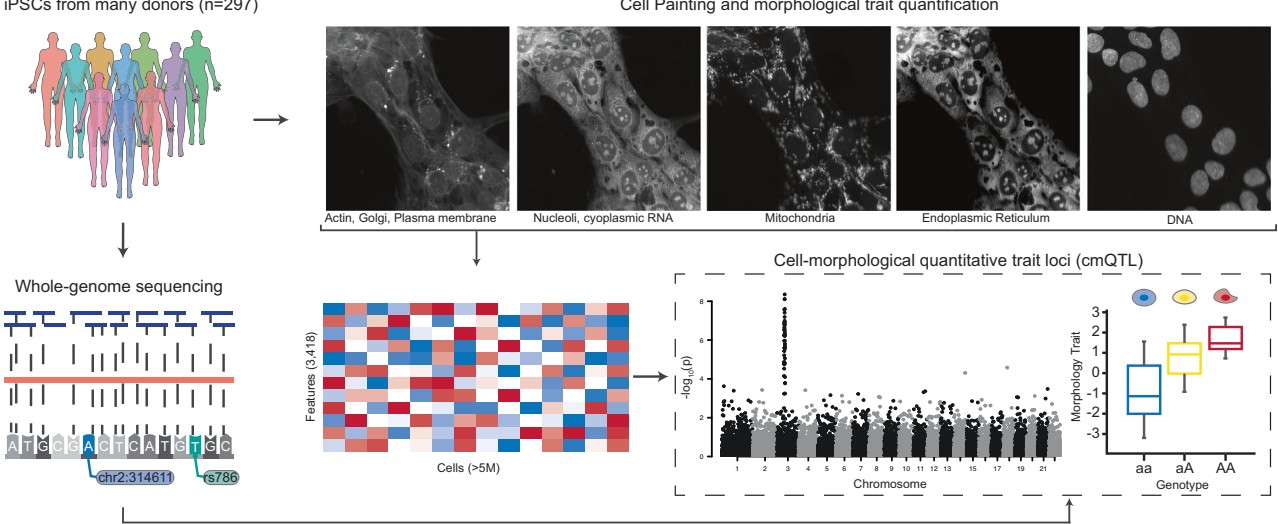

**Fig. 1 | Study overview.** iPSC lines from 297 donors were expanded, quality-control checked and then subject to both high-throughput imaging with Cell Painting and 30X whole-genome sequencing (WGS). Overall, we imaged 5.1 × 10⁶ cells across all donors and quantified 3418 morphological traits per cell using CellProfiler software. We inferred genetic variants from the WGS data and investigated whether individual morphological traits associated with both rare and common variation.

**Table 1 | Summary of donors' sex, disease status, age and tissue used for iPSC generation**

| Donor metadata | Value | |
|---|---|---|
| Sex | Male—52% ($n = 153$) | Female – 48% ($n = 144$) |
| Any disease | Yes—62% ($n = 184$) | No—38% ($n = 133$) |
| Age | 21 ± 10 | |
| Self-reported ancestry | European—70% ($n = 207$) | Non-European—30% ($n = 90$) |
| iPSC sample source | PBMCs—62% ($n = 184$) | Fibroblasts—38% ($n = 113$) |

could maximize differences between cell lines under these parameters (Fig. S1D). Each screening plate was stained with the standard Cell Painting dyes and imaged on a Perkin Elmer Phenix automated microscope within 48 h. We implemented this same workflow across all rounds of imaging.

Images were processed using CellProfiler to measure morphological traits and construct single-cell image-based profiles (Methods[21],). In total, we measured 3418 morphology traits for 5.1 million iPSCs from 297 donors after stringent QC (Methods, Fig. S2A, Table S2). We classified all morphological traits based on the cellular characteristics they represented, yielding five categories: Area and shape, Granularity, Intensity, Radial distribution, and Texture (Fig. 2a). Prior studies have shown that cells often displayed varied morphology in response to environmental cues and context[16]. To explore whether the contribution of genetic variation to cell morphology is context dependent, we segregated all cells into two groups based on whether they had any cells in contact (called colony cells, 97.48% of all cells) or not (called isolate cells, 2.52% of all cells) (Fig. S2B). We note that for the purposes of our study, "colony" refers to the number of neighbors a given cell has and is distinct from the colony terminology which is often used in basic stem cell culture practices.

We next performed 30X whole-genome sequencing (WGS) on all iPSC lines. Following quality control (QC, see Methods), we retained 7,020,633 common (minor allele frequency (MAF) > 5%) and 122,256 rare (MAF < 1%) variants for downstream analyses.

## Cell line characteristics and technical factors drive variability in morphological traits

Previous studies have shown that technical factors, including plate and well position can alter morphology-based readouts[22]. To explore the presence of cmQTLs in our data, we sought to identify technical factors which may confound our morphological phenotypes and remove these sources of variance from our downstream association tests. We performed a variance component analysis using well-level data to quantify the observed variance that can be attributed to each morphological trait by technical factors and cell line characteristics (Methods). We assessed the significance for each variance component, correcting for the number of tests, which was the product of the traits ($n = 3418$) and factors ($n = 9$) which include technical features such as imaging plate, well position, the number of cell neighbors, and whether the well was positioned on the edge of the plate (onEdge) in addition to demographic characteristics for the cell lines including genetic sex, reprogramming sample source, age of donor at the time of sample collection, and the clinical diagnosis for our tissue donors. We observed strong batch effects across imaging plates, which contributed the greatest degree of variance to our morphology traits (61.8 ± 17%, Fig. 2b, Fig. S3A). Several other confounders contributed varying levels of effect on different morphological traits (Fig. 2c). After correcting for these covariates, the remaining difference among cell line donors was significantly associated with all traits, explaining 16.7 ± 11% of the variance. (Fig. 2b). This indicated the potential for a genetic basis to the variability in morphology traits. Residual is the remaining (technical) variance in morphological traits which is

unexplained by the factors discussed above. Interestingly, the difference among donors explained a greater degree of variance in the trait category of AreaShape relative to the other trait categories (Wilcoxon rank sum test $P = 1.1 \times 10^{-55}$, Fig. S3B). We note that some of the shared variance may be explained by non-genetic factors, such as stable epigenetic modifications. We observed that many traits had very high pairwise correlation (*Pearson r* > 0.9) with one or more traits (Fig. S3C). To reduce redundancy in our downstream analyses, we selected a common set of 246 traits having $r < 0.9$ with each other by iteratively selecting a single representative trait for the set of correlated traits ($r > 0.9$) (Methods). We refer to this common set of 246 traits as "composite traits", which were used for our rare and common variant association tests (Table S3). We next summarized well-level morphology data into donor-level values (i.e., pseudo-bulk) by mean-averaging individual morphology traits across all wells for a given donor, resulting in one measurement per trait per donor ($N = 246$ traits and 297 donors) for both isolate and colony cells. These donor-level trait values were used for our quantitative association tests.

## Rare variant burden for cell morphological traits

Sequencing studies have identified hundreds of genes containing rare coding variants with association to disease burden[23–26]. These variants often have large effect sizes but explain a modest degree of total disease heritability[27]. To explore the effect of rare genetic variation on cellular morphology, we analyzed the association of composite traits ($n = 246$) with gene-level burden of protein-altering rare variants (MAF < 0.01). To ensure well-powered investigation, we examined 9105 genes that had rare variants in at least 2% of donors ($n >= 6$). We modeled individual morphology traits as a function of rare protein-altering variant burden in a gene, controlling for plate, well, and donor sex using linear regression (Methods). We performed our analysis separately for both colony and isolated cells. We identified 4 genome-wide significant associations between morphological traits and rare variant burden ($P < 2.2 \times 10^{-8}$, Bonferroni correction for 246 traits and 9105 genes) (Fig. 3a). These associations included one trait in colony cells and three traits in isolate cells. We did not observe any inflation in association statistics for these traits (Lambda ($\lambda$) = 1.01 for the association in colony cells and $\lambda = 1.01, 0.96, 0.98$ for the associations in isolate cells) (Fig. S4A). While the top feature associations (using a stringent Bonferroni correction) are quite different between isolate and colony cells, there is a modest overall correlation between the associations of morphology traits and genetic variants (Fig. S4B).

Rare variant burden in *WASF2* was negatively associated with a Zernike shape measure of the cytoplasm (*Cytoplasm_AreaShape_Zernike_9_3*) in colony cells ($n = 3$ missense and 1 in-frame deletion rare variants, β or effect size (se) = −1.24 (0.18), $P = 3.1 \times 10^{-10}$; Fig. 3b). Zernike features represent polynomial reconstructions of an organelle or object of cells. *WASF2* is named for its association with Wiskott-Aldrich syndrome, a rare genetic disorder which greatly increases the risk of various cancers ([28–30], and[31]). *WASF2* protein binds profilin, a G-actin-binding protein, promoting the exchange of ADP/ATP on actin and the formation of actin filament clusters[32,33]. The disruption of *WASF2* impairs actin formation and organization that could lead to their polarized distribution and spindle-shaped cells[34]. In representative images of cells with rare variants in *WASF2* it is difficult to identify this polarized and spindle-like shape by eye when compared to reference lines (Fig. S4C). In addition to our genome-wide association, rare variant burden in *WASF2* had nominal association ($P < 0.05$) with 90 other traits including 27 traits of area and shape category, suggesting *WASF2* may contribute to a range of cell morphological characteristics (Table S4).

Three traits were significantly associated with rare variant burden in the *PRLR* gene ($n = 6$ missense rare variants, β (se) = −1.17 (0.2), $P = 1.2 \times 10^{-8}$; Fig. 3c). The most interesting among these included asymmetries in the distribution of mitochondria in the perinuclear

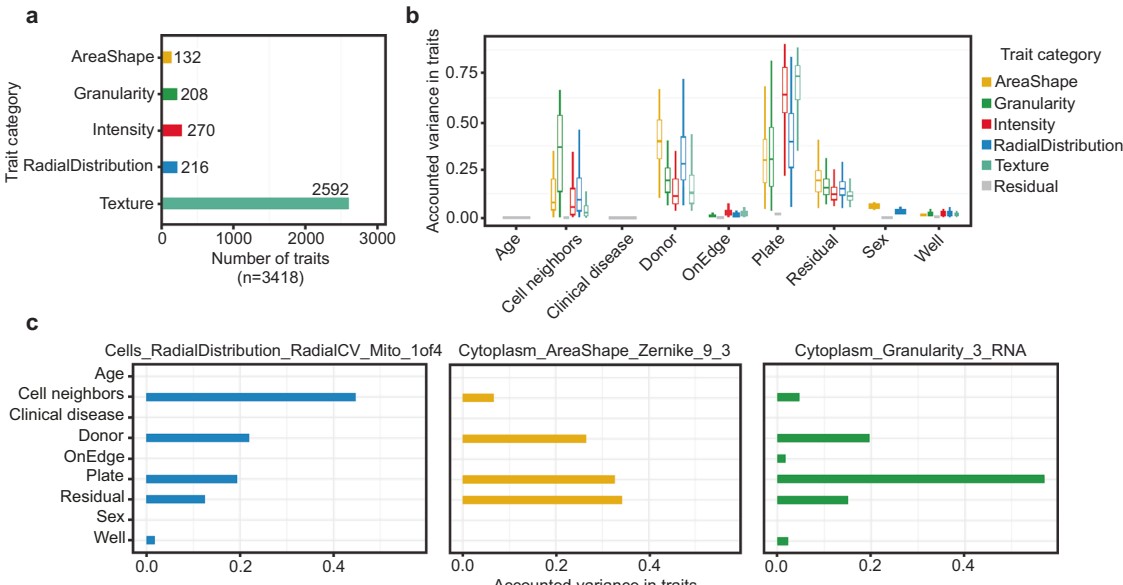

**Fig. 2 | Summary of morphological traits and variant component analysis.**
**a** Summary of five categories of morphological traits captured in our data
($n = 3418$). **b** Explained variance across all morphological traits ($n = 3418$). Data is
presented in a Tukey-style boxplot with the median (Q2) and the first and the
second quartiles (Q2, Q3) and error bars defined by the last data point within ±1.5-
times the interquartile range. **c** Exploring explained Variation in individual traits,
namely distribution of mitochondria around nucleus, cytoplasmic Zernike shape

metric 9_3, and cytoplasmic granularity in the RNA channel at scale 3, showed
differences in sources of variance, including technical effects such as plate and well
of imaging, whether the well was situated on the row or column on the edge of plate
(onEdge), biological sources such as donor. Donor ID represents the difference
among donors after accounting for their age, sex, disease-status and above-
mentioned imaging-related technical factors. Residual is the remaining unac-
counted variation in traits.

space (*Cells_RadialDistribution_RadialCV_Mito_1of4*). *PRLR* function has
been linked to several forms of cancers, including breast cancer and
lymphoma (Kavarthapu et al.[35], López Fontana et al.[36], Gharbaran
et al.[37]). *PRLR* encodes membrane-anchored receptors for a prolactin
ligand and is a part of the class-I cytokine receptor superfamily and
regulator of JAK-STAT5 pathway activity, regulating autocrine/para-
crine loops present in stem cells, which mediate their quiescence and
proliferation[38,39]. Previous findings in adipocytes showed *PRLR*
knockout alters mitochondrial packing and distribution throughout
the cell[40]. Moreover, rare variant burden in *PRLR* had nominal asso-
ciation ($P < 0.05$) with 118 other traits, providing more support to *PRLR*
as a genetic determinant of cellular morphology (Table S5).

We also inspected the associations with suggestive evidence, i.e.,
$P < 10^{-6}$. There were a total of 12 and 13 associations in colony and
isolated cells, respectively, which passed this threshold (Table S6). One
of the strongest associations in our suggestive results was between the
distribution in size of RNA particles in the cytoplasm (*Cytoplasm_-
Granularity_3_RNA*) and rare variant burden in *TSPAN15* gene ($n = 2$
missense and 1 splice region rare variants in the gene, β (se) = 0.9 (0.17),
$P = 3.7 \times 10^{-7}$; Fig. 3d). *TSPAN15* is expressed in all human tissues and
encodes for a cell surface protein[41]. A member of the tetraspanin family
of transmembrane segments, *TSPAN15* has been implicated in tumor-
related conditions (Huang et al.[42]). *TSPAN15* plays a role in cell activa-
tion and self-renewal through negative regulation of Notch-signaling[43].
Disruption of *TSPAN15* could lead to increased cell proliferation and
transcriptional activation which may be represented in our data by an
increase in the measurable RNA content in the cytoplasm.

To ensure our observed associations were not an artifact of our
nonparametric regression model, we permuted the data by randomly
assigning trait values across donors. These results suggested that our
observed significance of association between rare variant burden and
cell morphological traits was unlikely to have occurred by chance
(Fig. S4D). To confirm that our observed associations were not driven
by somatic variation introduced during iPSC reprogramming or those
which arise in cell culture (despite the short culture time in our study),

we repeated our association test while restricting to only those var-
iants that were previously observed in the gnomAD database (106,590
of 122,256 variants)[44]. All of our observed associations were recapitu-
lated (significant after Bonferroni correction for multiple testing and
with suggestive evidence) with concordant effect size and statistical
significance (*p*-value) (Fig. S4E). Taken together, our findings suggest
we could reliably identify associations between morphological traits
and rare protein-coding variants.

**Functional validation of rare variant associations**
CRISPR-based gene editing has been shown to be a viable mechanism
for validating gene expression phenotypes resulting from rare
variation[45]. To corroborate our rare-variant burden associations, we
examined whether knockdown of these genes impacted the same
morphological traits for which we identified a rare-variant burden
association. We transfected iPSCs from a single cell line expressing
constitutive dCas9-KRAB CRISPRi machinery with sgRNAs targeting
the transcriptional start site (TSS) for *WASF2*, *PRLR*, and *TSPAN15*
(Fig. 4a). Each gene was targeted by 2 different sgRNA sequences,
which were validated for knockdown of their target gene (25-95%
efficiency) (Fig. S5, Table S7). Cells transfected with non-targeting
sgRNAs were included as controls. We generated per-well (population-
averaged) morphological profiling using the same methods for our
discovery cohort. We compared the morphological trait values for our
rare-variant associations between non-targeting sgRNAs and those
targeting our genes of interest. For each gene tested, we observed the
predicted changes, and in the same direction, for each individual trait
relative to controls ($n = 28$ wells per targeting sgRNA and 52 wells per
non-targeting sgRNA, Welch's two sample *T* test, $P < 2.2 \times 10^{-16}$)
(Fig. 4b–d). Specifically, knockdown of *WASF2* resulted in a decreased
normalized score for the trait *Cytoplasm_AreaShape_Zernicke_9_3*
(Fig. 4b). We observed that a reduction in the expression of *TSPAN15*
coincided with an increase in trait score for *Cytoplasm_Granular-
ity_3_RNA* (Fig. 4d). Lastly, knockdown of *PRLR* decreased *Cells_Ra-
dialDistribution_RadialCV_Mito_1of4*, which defines the relationship

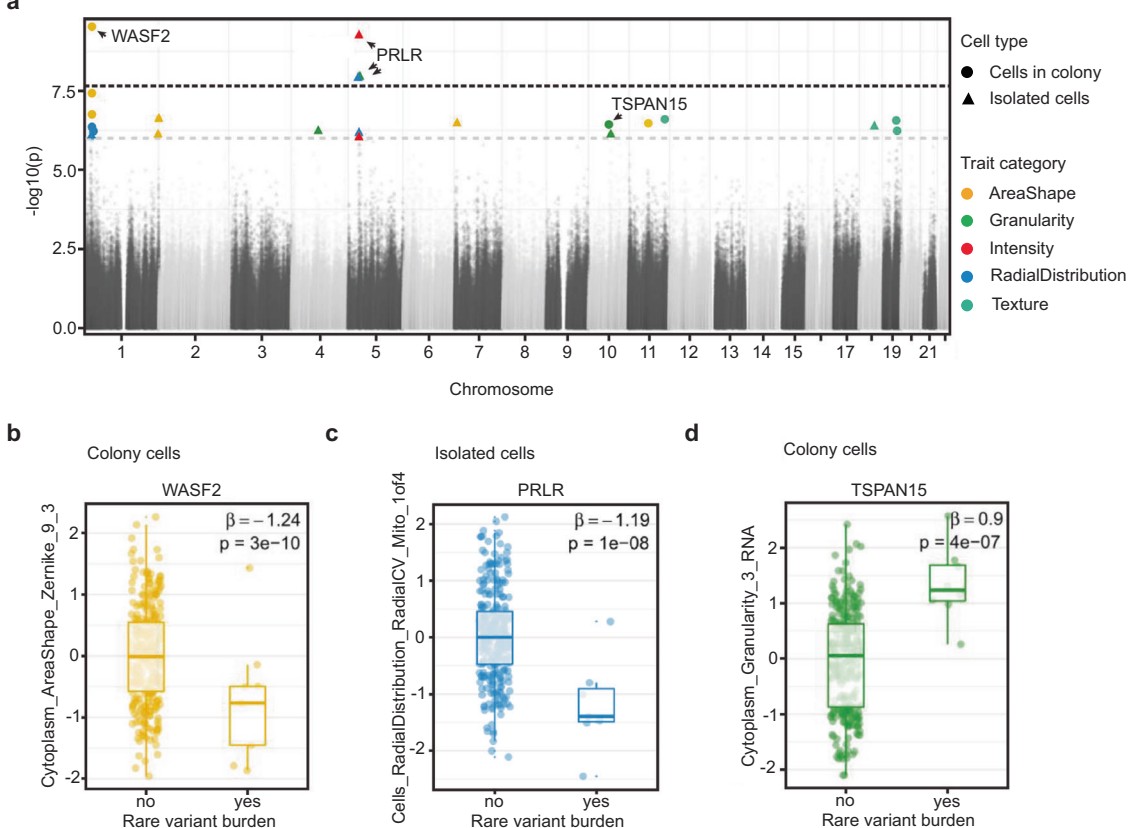

**Fig. 3 | Association between morphology and rare variant burden. a** Manhattan plot showing association between morphological traits ($n = 246$) and rare variant burden in candidate genes ($n = 9105$). Black dotted line represents the $p$-value threshold after Bonferroni correction for the number of tested traits and genes ($P = 0.05/246 \times 9105$, i.e. $2.2 \times 10^{-8}$). Gray dotted line represents the $p$-value threshold for suggestive evidence of association ($P = 10^{-6}$). **b**–**d** Box plots displaying the association between the Zernike_9_3 cytoplasm shape metric and rare variant burden in *WASF2* gene (**b**), distribution of mitochondria around the nucleus and rare variant burden in *PRLR* gene (**c**) and cytoplasmic granularity measure in the RNA channel and rare variant burden in *TSPAN15* gene (**d**). We provide the effect size (β estimate) and raw $p$-value of the association for each trait. Data is presented in a Tukey-style boxplot with the median (Q2) and the first and the second quartiles (Q2, Q3) and error bars defined by the last data point within ±1.5-times the inter-quartile range ($n = 297$ unique cell lines).

between the radial distribution of mitochondria around the nucleus (Fig. 4c). This effect is highlighted in representative images, whereby cells transfected with a *PRLR* targeting sgRNA displayed more uniform distribution of mitochondria around the nucleus when compared to non-targeting sgRNA cells where mitochondria tend to colocalize to one side of the nucleus (Fig. 4e).

**Common variant associations for cell morphological traits**
Genome-wide association studies (GWAS) have identified thousands of common variants that are associated with common diseases and traits. These variants have small effect sizes at the individual level but combine to explain a large degree of common disease heritability ([46,47], O'Connor et al.[27]). To identify common variants that are implicated in cell morphology, we performed 246 genome-wide association analyses, one for each composite trait. Each association was tested in colony and isolated cells separately (Fig. 5a, Fig. S6C). With our set of 297 donors, only one variant, rs315506, overlapping the chr17q11.2 locus, passed the genome-wide significance threshold (Bonferroni correction for 246 morphology traits, $5 \times 10^{-8}/246 = 2 \times 10^{-10}$). rs315506 is an intergenic variant and was associated with spatial distribution of endoplasmic reticulum (ER) in the cytoplasm (*Cytoplasm_RadialDistribution_RadialCV_ER_3of4*) in colonies (MAF = 0.08, β (se) = −0.52 (0.08), $P = 1.4 \times 10^{-10}$, Fig. S6A). This variant also showed suggestive evidence of association ($P < 10^{-5}/246 = 4.1 \times 10^{-8}$) with spatial distribution of ER near the periphery of cells (*Cells_RadialDistribution_Mean-Frac_ER_4of4*). rs315506 lies in the center of a 400 kb window

containing the genes *NF1*, *CORPS*, *UTP6* and *SUZ12*. Chromosomal alterations on chr17q11.2 cause NF1 microdeletion syndrome, which has been shown to impair protein localization to the ER[48,49]. To corroborate this observation, we analyzed the publicly available JUMP-Cell Painting data from U2OS cells that have perturbed *NF1* and *SUZ12* using CRISPR interference[50]. In this data, we see a significant change in our associated trait when NF1 and SUZ12 expression is decreased (Fig. 5b). In colony cells, the second strongest association was on chromosome 7 (between *Nuclei_Granularity_9_AGP* and rs36036340, MAF = 0.08, β (SE) = 0.38 (0.06), $P = 6 \times 10^{-10}$). rs36036340 lies within the gene *PRKAR1B*. Variants in *PRKAR1B* have been linked to neurodevelopmental disorders and activity of *PRKAR1B* has been shown to regulate tumorigenesis[51–53]. *PRKAR1B* mediates PI3K/AKT/mTOR pathway signaling through direct interactions between *PRKAR1B* and PI3K-110alpha[51]. We were unable to link perturbations in *PRKAR1B* to morphological changes for this feature using publicly available data (Fig. S6B). The most significant association in isolated cells was found on chromosome 13 (between *Nuclei_RadialDistribution_RadialCV_Brightfield_2of4* and rs9301897, MAF = 0.13, β (se) = −0.31 (0.05), $P = 4.5 \times 10^{-10}$) (Fig. S6C). rs9301897 lies within the gene *GPC6*, which is known to play a role in cell growth and division through the activation of cell surface receptors[54,55]. Over 300 loci reached the suggestive genome-wide significance threshold ($P < 4.1 \times 10^{-8}$, Table S8) indicating that a larger sample size and improved statistical power would be able to identify additional common variants associated with cell morphology. To confirm our observed associations were not attributable to noise, we permuted the

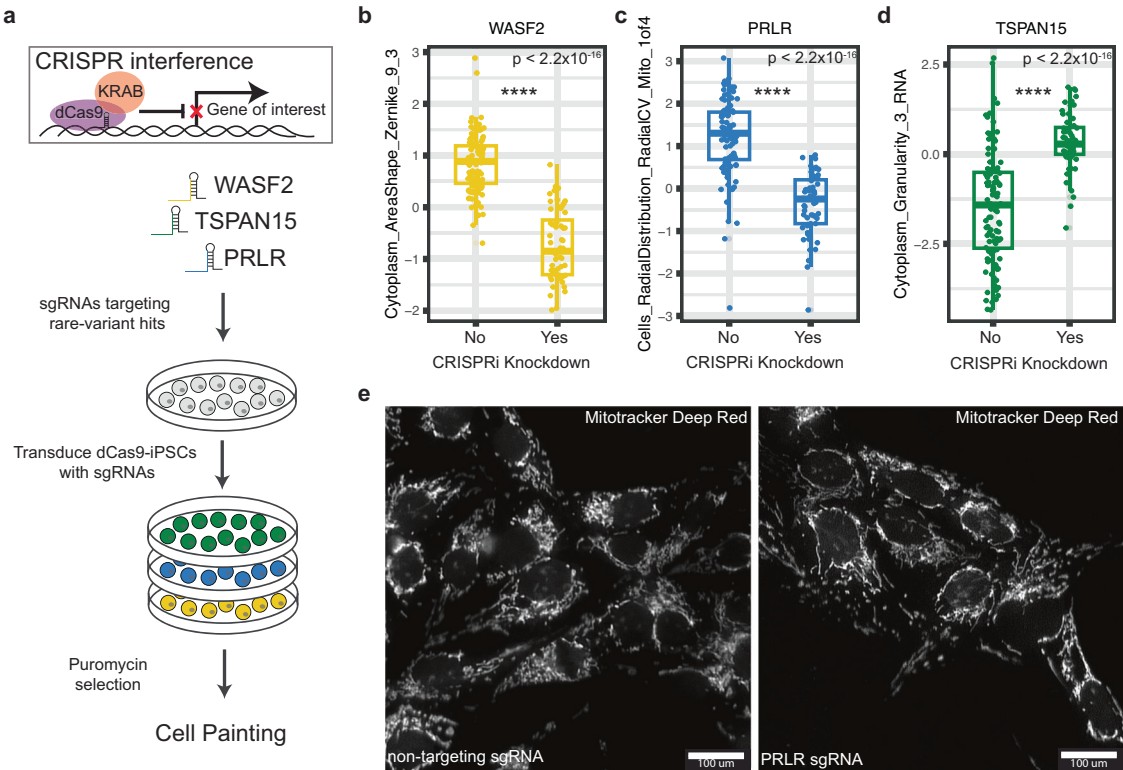

**Fig. 4 | Functional validation of rare-variant burden associations. a** Workflow for knockdown of rare-variant genes using CRISPR interference in iPSCs expressing constitutive dCas9-KRAB. **b–d** Box plots displaying quantification of traits between control non-targeting sgRNAs and sgRNAs targeting *WASF2*, *TSPAN15*, and *PRLR* on a per-well level *(n* = 52 wells per non-targeting sgRNAs*, n* = 56 wells per targeting sgRNAs*, P* < 2.2 × 10⁻¹⁶, Welch's two-sided *T* test*)*. Effect on the trait score is consistent with what we observed in our rare-variant burden association. Data is presented in a Tukey-style boxplot with the median (Q2) and the first and the second quartiles (Q2, Q3) and error bars defined by the last data point within ±1.5-times the interquartile range. **e** Representative image of an observable gene-trait association for *PRLR*. Cells_RadialDistribution_RadialCV_Mito_1of4 relates to the asymmetric distribution of mitochondria in the ring right around the nucleus. In the non-targeting controls (left) we observed clustering of mitochondria on a particular side of the nucleus, whereas in the *PRLR* knockdown sgRNA (right) we observed a more distributed presence of mitochondria around the nucleus.

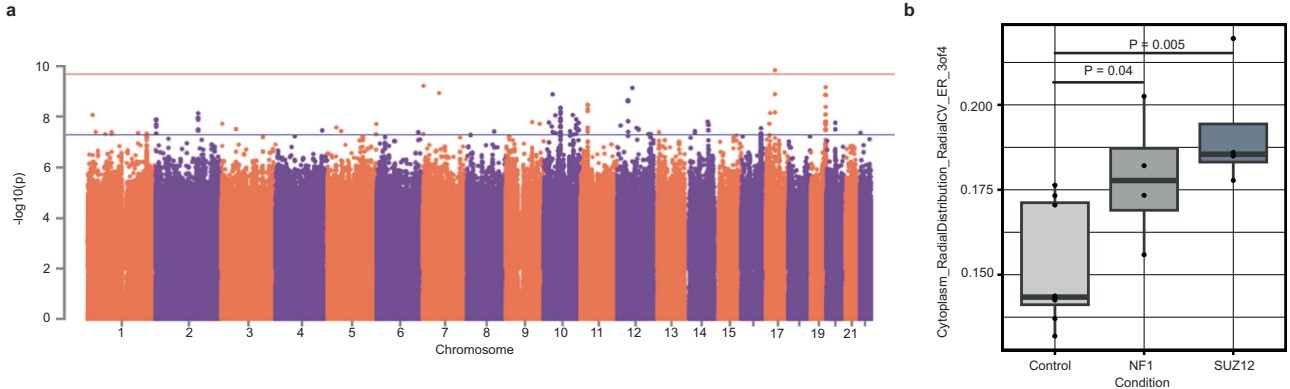

**Fig. 5 | Common variant analysis. a** Manhattan plot for trait association test in colony cells. Red line represents the *p*-value threshold after Bonferroni correction for the number of tested traits and genes (*P* < 2 × 10⁻¹⁰) Blue line represents the *p*-value threshold for hits with suggestive evidence (*P* < 4.1 × 10⁻⁸). **b** Impact of knockdown of *NF1* and *SUZ12* in U2OS cells on *Cytoplasm_RadialDistrubtion_RadialCV_ER_3of4*

(*P* = 0.04 *NF1*, *P* = 0.005 *SUZ12*, Welch's two-sided *T* test). Data are presented in a Tukey-style boxplot with the median (Q2) and the first and the second quartiles (Q2, Q3) and error bars defined by the last data point within ±1.5-times the interquartile range (*n* = 8 control biological replicate, 4 biological replicates for each target gene).

data by shuffling genotype labels and repeating the association tests. These results suggested that our observed significance of association between common variants and cell morphological traits was unlikely to have occurred by chance (Fig. S6). Moreover, several loci (Table S8) showed suggestive association with more than one trait suggesting shared genetic etiology among different morphological traits.

## Sample size requirements and predictions for future cellular genetic studies

There has been an emergence of cellular genetics studies that aim to uncover the biological function of genetic variation (Wolter et al[56,57]., Miller[58].). When compared to genetics studies of quantitative traits, cellular GWAS are limited in sample sizes, which provide a barrier to

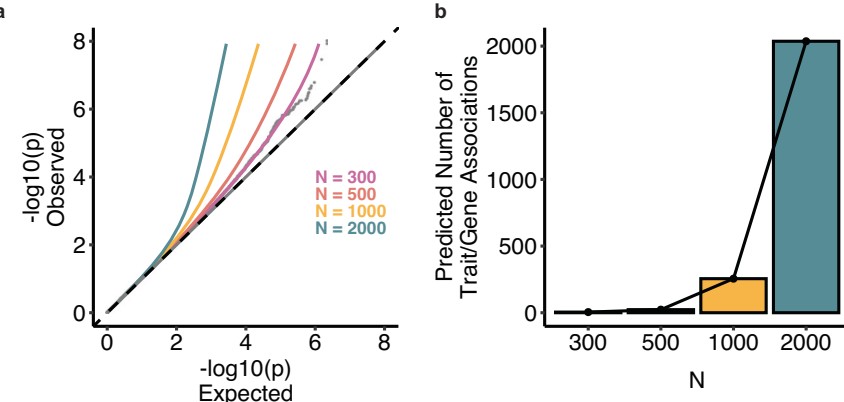

**Fig. 6 | Distribution of effect sizes and predictions for future discovery. a** QQ plot with current observed tests, model fit, and three predicted lines corresponding to $N = 500, 1000, 2000$. **b** Barplots showing the number of discoveries per trait at current $N$, then projected $N = 500, 1000, 2000$.

the discovery of significant associations. We sought to understand the power of our study with a sample size of nearly 300 individuals, and a distribution of effect sizes spanning common and rare variants. Our findings suggest that genetic discovery for cell morphological phenotypes was achievable at our current sample size. However, the small number of significant discoveries in our analysis begs the question of how many discoveries can be made at larger sample sizes. If the discovery of many hits required a few thousand samples, such experiments would be feasible and worthwhile; if such discovery required hundreds of thousands to millions of samples, it may be out of reach for the foreseeable future. To answer this question, we estimated the distribution of common and rare variant effect sizes using Fourier Mixture Regression noLD (FMR-noLD) (Methods; O'Connor[59]). Briefly, FMR-noLD fits a flexible mixture model to the distribution of effect sizes. This mixture model can be used to simulate effect sizes at various sample sizes, predicting how many significant discoveries will be made.

For common variants, we analyzed summary statistics from a pruned set of approximately 350,000 variants (Methods). We found that our dataset was underpowered for this analysis: FMR-noLD inferred that essentially all common variant effect sizes are 0, which is implausible and the expected behavior in the low power regime. For rare variants, we analyzed summary statistics from the main burden association analysis described earlier. In contrast to the common variant analysis, we predict that many discoveries will be made at feasible larger sample sizes, with more than 250 significant discoveries at $N = 1000$ and more than 2000 discoveries at $N = 2000$ (Fig. 6a, b). Overall, our study was underpowered to detect a significant number of associations between common haplotypes and cell morphological traits, but our rare variant analyses provide a promising path for future studies exploring the impact of rare genetic variation on cell morphology.

## Discussion

Previous studies linking genetic variants to cellular function have largely focused on human genes and alleles which mediate molecular phenotypes, such as gene or protein expression and chromatin accessibility[8,56,57,60]. To expand on these studies, we combined high-throughput cell culture techniques with cost-effective and high-dimensional image-based cell profiling (i.e., Cell Painting) to link genetic variants to their morphological function in 297 donors.

Our work provides the largest to date exploration of genetic influences on cell morphology (what we term cmQTLs). Where previous studies have been limited by both sample size and the scale of morphological measurements, we combined whole genome sequence analysis with Cell Painting to define relationships between genetic

variants and morphological traits extracted from >5 M iPSCs. We identified confounding factors that drive variation in cellular phenotypes which are important to address when performing similar studies. In particular, attenuating batch effects across plates and well position is critical in imaging-based assays. To address this challenge, we incorporated automated liquid handling devices to maximize plate distribution of cell lines across 384 well microplates.

We measured associations between rare variant burden and morphological traits, identifying novel associations between *WASF2*, *PRLR*, *TSPAN15* and morphological phenotypes related to cytoplasmic area and shape, nucleic granularity, and the distribution of mitochondria around the nucleus. These associations were validated by CRISPR-mediated knockdown and supported by mechanistic information about these genes from the literature. Even though our effective knockdown had a range of efficiency (25-95%) we were able to measure meaningful changes in morphological features even at the lower range. This is consistent with previous work showing that gene expression is often stochastic and subtle changes in expression may lead to large changes in functional protein[61]. Each of the genes nominated in our rare variant analysis has been implicated in various cancers. We find this result interesting and somewhat unsurprising, given that pluripotent stem cells exhibit self-renewal properties which closely resemble cancer cells. These results suggest that genetic studies in iPSCs may shed meaningful insights into cancer-linked genes. It will be important for future studies to measure whether these associations are cell-type specific, or if they would be retained using differentiated cells. We extended our analysis to look for associations between morphological traits and common haplotypes. We found one significant result and 300 potential associations. We corroborated our significant common variant association with publicly available data showing that perturbations of nearby genes impact the associated morphological trait.

Interestingly, we observed no overlap in traits and associated variants between colony and isolated cells, suggesting a differential effect of genetic variation based on the environmental context of the cells. This is consistent with previous studies that have shown that intrinsic properties of cells may only come to light in the context of altering the cellular environment[16,62,63]. In this study, we pseudo-bulked single-cell profiling data to generate per-donor trait scores. It will be interesting for future work to examine morphology at the single-cell level, similar to single-cell RNA sequencing approaches to better understand genetic influences on cellular heterogeneity.

The small number of significant discoveries in our work highlights that in vitro genetic studies still require substantial increases in sample sizes to saturate discovery potential. Our common variant analysis suggests that we are vastly underpowered to measure genetic

associations to cell morphology and our estimated effect size distributions infer that cell morphology may behave similarly to quantitative traits. As discovery potential for quantitative traits often scales linearly with the number of samples included in the study, our data suggests that even with 3000 genetically unique cell lines, we may still only yield 10 genome-wide significant common variant cmQTLs. This is a sobering result, as it suggests that tens of thousands of cell lines would be needed to begin mapping SNP-trait associations for cell morphology. In contrast, our analysis of rare variant effect sizes suggests that with modest increases in sample sizes, we are well-positioned to detect many rare variant cmQTLs. Future studies which can leverage 1000-2000 unique cell lines may yield many 1000 s of genome-wide significant gene-trait associations. While scaling in vitro studies to 2000 cell lines will still be a large hurdle, it is one which can be feasibly overcome with current iPSC collections.

Our work has several limitations that highlight directions for future research. This study focused on how germline genetic variation influences stem cell morphology. While we applied a rigorous quality control to identify and remove cells with abnormal karyotypes or large genomic rearrangements, it is possible that new somatic mutations may arise over time in culture. It will be important for additional studies to explore how recurring somatic mutations mediate cell morphology. Furthermore, the cell types utilized in this study are in a basal, undifferentiated state. It will be valuable for future studies to explore these associations in more physiologically relevant contexts, where disease-associated variants are enriched[64]. These findings suggest this framework could be applied to relevant cells and tissues such as iPSC-derived differentiated cells, post-mortem brain samples or excisable somatic cells. Moreover, we did not find any cell morphological traits associated with clinical disease categories from the cell line donors (data not shown). Similar to exploring common variant associations, we are likely underpowered in any single disease category to identify significant associations. There have been many studies elucidating morphological features associated with various diseases, but they often contained larger sample sizes and incorporated more specialized cell types[22,65,66]. Extending our current study to diverse cell types and increasing the number of samples for clinical disease categories will be a critical next step in efforts to link cell morphology to human illnesses.

This approach holds significant promise for future studies leveraging human-derived, disease-relevant cell types for modeling the impact of genetic variation on cellular function. The use of imaging to capture phenotypes is particularly attractive in experimental designs for several reasons, such as the low cost per cell for imaging, and the ease of processing data and preparation of the cells or tissues as compared to the generation of other molecular data such as RNA-sequencing or epigenetic assays[67]. Moreover, large imaging datasets provide tools for developing robust statistical models for combined analysis of morphological profiling data with additional modalities such as gene expression to comprehensively interrogate genetic variants and their function[68]. Taken together, we demonstrate cellular morphology can be a cost-effective readout for modeling the biological function of human genetic variation.

## Methods
### Materials and iPSC generation
Our dataset comprised 297 donors from the iPSC repository of California Institute for Regenerative Medicine (CIRM) (Lin et al.[69]) (Table S1). All cell lines are available from CIRM. Either B cells or Fibroblasts were taken from each donor from which iPSC lines were generated using non-integrating episomal vectors[70]. The cell lines included in this work contained both male and female cell lines (52% & 48%, respectively), between the ages of 1 to 90 (Table S1). While the majority of the samples were derived from individuals of European ancestry, the cohort also included cell lines from individuals of East Asian, African, and Admixed ancestry (Fig. S2, Table S1). Cell line growth rates were calculated by measuring the number of cells plated during their final expansion and the number of cells measured prior to cryobanking. The doubling time was calculated by comparing the increase in cell number relative to the number of hours in culture. Each iPSC sample underwent Global Screening Array (GSA) for karyotype analysis to ensure chromosomal integrity, as well as 30X whole-genome sequencing to determine genome-wide variants for each donor. Any cell lines displaying abnormal karyotypes or genomic rearrangements > 1 Mb were excluded from our study. Each iPSC line was cultured between passages 12 and 15 before use in the experiment.

### iPSC culture
Human iPSCs were maintained on plates coated with geltrex (life technologies, A1413301) in StemFlex media (Gibco, A3349401) and passaged with accutase (Gibco, A11105). All cell cultures were maintained at 37 °C, 5% CO2.

### Cell seeding and staining
For each batch of imaging, cells were detached from 6-well NUNC plates using Accutase (StemcellTech; cat#07920) for generating single-cell suspensions. Following detachment, cells were centrifuged at 1000 rpm × 5:00 and re-suspended in StemFlex medium supplemented with 10uM ROCK inhibitor. After each cell line was counted to determine cell solution concentration and viability, the desired cell solution volume was aliquoted into a 96-deep well low attachment plate following a specific plate map to ensure that wells from any given cell line were not predominantly on the edge wells or too close together. To disperse a high number of cell lines across a 384-well plate in a semi-random fashion, we optimized the use of an Agilent Bravo liquid handling device (Fig. S1B). Here, using an 8-channel head, cell solutions were transferred from the 96-well low attachment plate and distributed into a geltrex-coated Perkin Elmer Cell Carrier 384-well plate at a density of 10,000 cells per well. Each cell line was plated into 8 distinct wells on the final screening plate in four-well quadrants (see Fig. S2). These parameters were selected based on a pilot experiment with 6 cell lines across a range of densities and fixation conditions. We observed that we could maximize variability across cell lines using 10k cells per well fixed 6 h after plating, when compared to 24 h post-plating.

### Cell painting and imaging
Cells were stained and imaged with minor adaptations to procedures described previously[17,18]. Six hours post seeding in 384-well plates, cells were treated for 30 min with 0.5 uM MitoTracker Deep Red FM - Special Packaging (Thermo Fisher cat#: M22426) dye at 37ºC. Following the MitoTracker treatment, cells were fixed with 16% paraformaldehyde diluted to a final concentration of 4% (Thermo Fisher cat#: 043368.9 M) for 20 min in the dark at RT. After three washes with 1X HBSS cells were permeabilized and stained using a solution of 1X HBSS (Thermo Fisher cat#: 14175095), 0.1% Triton-X-100 (Sigma Aldrich cat#: X100-5ML), 1% Bovine Serum Albumin, 8.25 nM Alexa Fluor 568 Phalloidin (Thermo Fisher cat#: A12380), 0.005 mg/ml Concanavalin A, Alexa Fluor 488 Conjugate (Thermo Fisher cat#: C11252), 1ug/ml Hoechst 33342, Trihydrochloride, Trihydrate (Thermo Fisher cat#: H3570), 6uM SYTO 14 Green Fluorescent Nucleic Acid Stain (Thermo Fisher cat#: S7576), and 1.5ug/ml Wheat Germ Agglutinin, Alexa Fluor 555 Conjugate (Thermo Fisher cat#: W32464) for 1 h at RT in the dark. Following the staining, plates were washed 3X with 1X HBSS and sealed until imaging. Cell Painted plates were imaged on a Perkin Elmer Phenix Automated Microscope under a standardized protocol[17]. Configuration files for imaging protocols can be found with their associated images at https://registry.opendata.aws/cellpainting-gallery/ under project ID "cpg0022-cmqtl." All 297 cell lines were dispersed across seven plates which were imaged in four separate batches.

## Quantification of cellular morphology traits and their quality control

The segmentation of individual cells in the image into their cellular compartments (whole cell, cytoplasm and nuclei) and subsequently quantification of morphology traits for each cellular compartments was done using CellProfiler 3.1.8 (McQuinn et al. 2018); pipelines are available at https://github.com/broadinstitute/imaging-platform-pipelines/tree/master/cellpainting_ipsc_20x_phenix_with_bf_bin1. Analysis of CRISPR experiments was done in CellProfiler 4.2.4 with pipelines available at https://github.com/broadinstitute/imaging-platform-pipelines/tree/master/cellpainting_ipsc_20x_phenix_with_bf_bin1_cp4[71]. Subsequently, cells missing measurement for more than 5% of traits were removed. Morphology traits a priori known to be problematic, not measured across all cells or non-variable across cells were removed using Caret v6.0-86 package. QC-ed cells were then segregated into two groups based on the number of neighbors: isolated cells having no neighbors and colony cells having one or more neighbors. Individual morphology traits were then summarized to well level measurement by averaging them across all cells per well, resulting in a well by trait matrix. Following this, each morphology trait was gaussianized across all 7 plates using inverse normal transformation (INT) method.

## Selection of traits for association analysis

A set of morphology traits for association analysis (with both common variants and rare variant burden) was selected by considering their pairwise correlation across colony and isolate cells in the following steps: Step 1. Calculate Pearson correlation matrix for colony and isolate cells at donor level (total 2 correlation matrices). Step 2. Identify that single trait having the *Pearson r* ≥ 0.9 with the largest number of other traits across both correlation matrices. We specifically chose *Pearson r* ≥ 0.9 as cutoff here because most traits (93.7% and 91.2% traits in colony and isolated cells, respectively) had a correlation *Pearson* ≥ 0.9 with at least one other trait (Fig S7). Step 3. Include that individual trait for association analysis. Remove it and other traits having *Pearson* ≥ 0.9 with it from correlation matrices. Step 4. Repeat steps 1 to 3 until there are no more traits to include in the association analysis.

## Whole genome sequencing (WGS), variant calling and genes to test

DNA was obtained from cell line pellets with the Qiagen Quick-Start DNeasy Blood and Tissue Kit (cat. no. 69506). DNA samples were submitted to the Genomics Platform at the Broad Institute of MIT and Harvard. Whole genome sequencing (30x) was performed for all individuals ($n = 297$) at the Broad Institute Genomics Platform using Illumina Nextera library preparation, quality control, and sequencing on the Illumina HiSeq 2500 platform. Raw sequences were QC-ed and sequencing reads (150 bp, paired-end) were aligned to the hg38 reference genome using the BWA alignment program. Variants were called and annotated (VQSLOD filter) using HapMap reference.

## WGS data quality control for common variant association analysis

The QC-ed WGS VCF file was processed using plink v1.90b3 to remove sex chromosomes, multi-allelic variants, variants with duplicated positions, and small insertions and deletions larger than 5 bp. Of 38,239,223 variants loaded from the VCF file, 33,348,914 passed these filters. Donor-level genotype missingness rates were checked to exclude donors with genotype missingness rates > 10%. All 297 individuals passed this filter. Finally, variants with minor allele frequency (MAF) < 5%, missingness > 5%, and Hardy-Weinberg equilibrium *p*-value < $10^{-5}$ were excluded, following which, 7,020,633 remained for common variant association analysis.

## Principal components analysis (PCA)

Plink v1.90b3 was used on common (MAF > 5%) and post-QC variants to remove regions with known long-range linkage disequilibrium (LD) and variants in high LD (r2 > 0.1 in a window of 50 kb and a sliding window of 10 kb) (Price A. L. Am. J. Hum. Genetics 2008). The remaining 291,493 variants were loaded to GCTA v1.91.1 to generate a genetic relatedness matrix (GRM) using the --make-grm command with default options. The resulting GRM was used to generate 20 PCs using GCTA v1.91.1 --pca command with default options.

## Variance component analysis

Variance component of fixed (cell neighbor density and donor's age) and random effects (iPSC source tissue, cell line ID, plate and well of imaging, donor's sex, and disease status) was estimated for selected traits using linear mixed model (lmer function in lmertest package). We included the first 4 PCs derived from genetic variation, corresponding to the elbow in scree plot, for ancestry/population stratification. The *p*-value of each factor was Bonferroni corrected for the number of all tested traits ($n = 3418$).

Linear model question for variance component analysis:

$$\text{Gaussianized trait} \sim (1|\text{Disease}_{\text{yes}|\text{no}}) + (1|\text{iPSC}_{\text{fibroblast}|\text{Bcell}})$$
$$+ (1|\text{Sex}_{\text{male}|\text{female}}) + \text{Age} + \sum PC_{i=1-4} + (1|\text{Plate}) + (1|\text{Well})$$
$$+ (1|\text{OnEdge}_{\text{yes}|\text{no}}) + (1|\text{Cell line ID}) + \text{Neighbor count}_{\text{ifnotisolets}}$$

## Common variant association analysis

The linear regression framework implemented in GCTA v1.91.1 (--fastGWA-lr command) was used to test the association of common (MAF > 5%), post-QC variants with 246 post-QC, INT traits that were described above. Like the rare variant association analysis, plate and sex were included as categorical and four genotyping PCs, number of cell neighbors (for cells in colony) and the edge variable were included as quantitative variables in the model. Associations were considered statistically significant if they passed the genome-wide significance threshold for 246 tests ($P < 5 \times 10^{-8}/246$).

Linear model equation for isolate cells:

$$\text{Gaussianized trait} \sim \text{Variant} + \text{Age} + \text{Sex} + \sum PC_{i=1-4} + (1|\text{Plate})$$

Linear model equation for colony cells:

$$\text{Gaussianized trait} \sim \text{Variant} + \text{Age} + \text{Sex} + \sum PC_{i=1-4} + (1|\text{Plate})$$
$$+ \text{Neighbor count}$$

## Rare variant burden test

To perform the rare variant burden test, the variants which were autosomal, passed the VQSLOD filter and called in >95% individuals and had maf<1% were retained. These variants were annotated for their functional effect using SnpEff v5.0. After annotation, those variants were kept which resided in the protein-coding region and had high or moderate effects on encoded protein. For each gene, multiple rare variants were grouped and coded as present or absent. The association between individual morphology traits and the presence of rare variants in a gene was investigated using linear regression models. The *p*-values of associations were corrected for both the number of tested traits ($n = 246$) and the number of genes ($n = 9105$) using the Bonferroni correction method.

## CRISPRi sgRNA design, cloning, and virus production

To functionally validate the rare-variant burden associations, we designed sgRNAs targeting the transcriptional start site (TSS) for each gene using CRISPick software (Doench, 2016, Sanson, 2018). sgRNA

oligonucleotides were cloned into the CROPseq vector using a Golden Gate cloning protocol (Addgene: #106280, Juong, 2017). To validate sequence insertion, DNA plasmids were sequenced by a 3rd party provider. Plasmids with successful insertion were packaged for lentivirus generation using *Trans*IT-293 reagent (Mirus Bio cat#: MIR 2704) and packaging plasmids VSV-G and DVPR (Addgene: #12259 and #12259).). HEK239T (ATCC cat#: CRL-3216) cells were transfected with sgRNA packaging plasmid and incubated for 48 h. HEK239T media supernatant was collected, and lentivirus was concentrated using LENTI-X concentrator (Takara) per the manufacturer's instructions. The virus supernatant was then aliquoted and stored at -80C.

### sgRNA transduction in dCas9-iPSCs
An iPSC line, WTC11_TO-NGN2_dCas9-BFP-KRAB (gift from Michael Ward), was seeded at 250k cells per well in a 12-well plate and 50ul of sgRNA lentivirus was added to each designated well. This iPSC line was cultured using mTeSR1 medium according to source recommendations (Stemcell Technologies, cat#: 85850). The following day, 1 mL of mTeSR1 complete media was added on top of the existing media. 48 h post-transduction, cells underwent a full media change with the addition of 1 ug/ml puromycin (Sigma Aldrich cat#: P8833) for chemical selection of cells which did not uptake the sgRNAs. Puromycin is supplemented in the feeding media for the duration the cell line is in culture to avoid uninfected cells from populating the dish.

### qPCR analysis
RNA isolation was performed with the Direct-Zol RNA miniprep kit (ZYMO: cat# R2051) according to the manufacturer's instructions. To prevent DNA contamination, RNA was treated with DNase I (ZYMO: cat# R2051). The yield of RNA was determined with a Denovix DS-11 Series Spectrophotometer (Denovix). 200 ng of RNA was reverse transcribed with the iScript cDNA Synthesis Kit (Bio-Rad, cat# 1708890). For all analyses, RT–qPCR was carried out with iQ SYBR Green Supermix (Bio-Rad, cat# 1708880) and specific primers for each gene (listed below) with a CFX384 Touch Real-Time PCR Detection System (Bio-Rad). Target genes were normalized to the geometric mean of control genes, *RPL10* and *GAPDH*, and relative expression compared to the mean Ct values for non-targeting control sgRNAs and gene targeting sgRNAs, respectively.

The following primers were used:
WASF2_forward 5'-TAGTAACGAGGAACATCGAGCC-3'
WASF2_reverse 5'-AAGGGAGCTTACCCGAGAGG-3'
PRLR_forward 5'-TCTCCACCTACCCTGATTGAC-3'
PRLR_reverse 5'-CGAACCTGGACAAGGTATTTCTG-3'
TSPAN15_forward 5'-TCCCTCCGTGACAACCTGTA-3'
TSPAN15_reverse 5'-CCGCCACAGCACTTGAACT-3'
RPL10_forward 5'-GCCGTACCCAAAGTCTCGC-3'
RPL10_reverse 5'-CACAAAGCGGAAACTCATCCA-3'
GAPDH_forward 5'-GGAGCGAGATCCCTCCAAAAT-3'
GAPDH_reverse 5'-GGCTGTTGTCATACTTCTCATGG-3'

### Modeling cmQTL effect size distributions with FMR-noLD
We used FMR-noLD (O'Connor, 2021) to model the effect size distribution for both common and rare variant summary statistics from our analyses. FMR-noLD is a simplified version of the main FMR method that does not model linkage disequilibrium (LD) between variants. We used FMR-noLD rather than FMR for this analysis as 1) the mixed ancestry of our sample complicates LD-score style estimators such as FMR, and 2) FMR-noLD is the appropriate choice for rare variants, which have very little LD. For the common-variant analysis, we used the PLINK2[72] –indep-pairwise (with parameters: variant count 50, variant count shift 5, threshold 0.2) utility to find a set of approximately 350,000 variants in approximate linkage equilibrium. We then submitted the concatenated set of summary statistics across all traits for FMR-noLD. For the rare-variant analysis, we used the same set of summary statistics used in the main burden test analysis (i.e. with no need for variant pruning), concatenated across all traits. For power analysis for rare variant association, we first predicted effect size distributions at varying sample sizes by adding sampling variance $1/N$ to our inferred distribution of true effect sizes. We then computed the cumulative distribution function of these predicted distributions at our significance threshold for the main rare variant burden analysis, $p = 2.2e-8$, which represents the proportion of tests that are expected to be significant at each sample size.

### Reporting summary
Further information on research design is available in the Nature Portfolio Reporting Summary linked to this article.

## Data availability
The whole genome sequence data generated in this study has been deposited in the NCBI dbGaP database under accession code phs002032.v1.p1. https://www.ncbi.nlm.nih.gov/projects/gap/cgi-bin/study.cgi?study_id=phs002032.v1.p1. The raw image data are available in the Cell Painting Gallery on the Registry of Open Data on AWS (https://registry.opendata.aws/cellpainting-gallery/) as dataset 'cpg0022-cmqtl' at no cost and no need for registration.

## Code availability
Source code to reproduce and build upon the presented results is available at https://github.com/broadinstitute/cmQTL.

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

## Acknowledgements

The authors thank members of the Nehme, Carpenter-Singh, and Raychaudhuri labs for insightful discussions and critical reading of the manuscript, and Anna Neumann for project management. This work was supported by a Broad Institute Variant to Function (V2F) Initiative grant, the National Institutes of Health (NIMH U01 MH115727 to RN/KE/SAM and NIGMS MIRA R35 GM122547 to AEC), the Chan Zuckerberg Initiative DAF, an advised fund of the Silicon Valley Community Foundation (grant number 2020-225720 to BAC), as well as the Stanley Center for Psychiatric Research at the Broad Institute. The authors also gratefully acknowledge the use of the PerkinElmer Opera Phenix High-Content/High-Throughput imaging system at the Broad Institute, funded by the S10 Grant NIH OD-026839-01.

## Author contributions

M.T., J.A., S.A., A.E.C., S.S., R.N. and S.R. conceived the work and wrote the manuscript. R.N. obtained cell lines from CIRM and coordinated the sequencing. M.T. with support from E.P. and D.L. performed cell culture data generation and CRISPRi knockdown validation. J.A., S.A. performed genetic association analyses. B.A.C., S.S., G.W., M.H., E.W. processed and analyzed Cell Painting data. A.E.C., S.S., R.N. and S.R. supervised the work and analyses.

## Competing interests

The authors declare no competing interests.

## Additional information

[1]Stanley Center for Psychiatric Research, Broad Institute of MIT and Harvard, Cambridge, MA, USA. [2]Department of Stem Cell and Regenerative Biology, Harvard University, Cambridge, MA, USA. [3]Centre for Gene Therapy and Regenerative Medicine, King's College, London, UK. [4]Center for Data Sciences, Brigham and Women's Hospital and Harvard Medical School, Boston, MA, USA. [5]Division of Rheumatology, Inflammation, and Immunity, Department of Medicine, Brigham and Women's Hospital and Harvard Medical School, Boston, MA, USA. [6]Division of Genetics, Department of Medicine, Brigham and Women's Hospital and Harvard Medical School, Boston, MA, USA. [7]Program in Medical and Population Genetics, Broad Institute of MIT and Harvard, Cambridge, MA, USA. [8]Department of Biomedical Informatics, Harvard Medical School, Boston, MA, USA. [9]Institute for Genomic Health, Icahn School of Medicine at Mount Sinai, New York, NY, USA. [10]Imaging Platform, Broad Institute of MIT and Harvard, Cambridge, MA, USA. [11]Analytic and Translational Genetics Unit, Massachusetts General Hospital, Boston, MA, USA. [12]Department of Epidemiology, Harvard T.H. Chan School of Public Health, Boston, MA, USA. [13]Halıcıoğlu Data Science Institute, University of California, La Jolla, CA, USA. [14]Department of Medicine, University of California, La Jolla, CA, USA. [15]Department of Genetics, Harvard Medical School, Boston, MA, USA. [16]Centre for Genetics and Genomics Versus Arthritis, Manchester Academic Health Science Centre, University of Manchester, Manchester, UK. [17]These authors contributed equally: Matthew Tegtmeyer, Jatin Arora, Samira Asgari. ✉e-mail: shsingh@broadinstitute.org; rnehme@broadinstitute.org; soumya@broadinstitute.org

