## [Peer Review File · Nature Communications]

Reviewers' Comments:

Reviewer #1:

Remarks to the Author:

Tegtmeyer et al. conducted a cell morphological quantitative trait loci analysis investigating the associations in 297 iPSC lines from different donors between SNPs and 3,000 cellular morphological traits obtained using high-throughput imaging of multiple cellular compartments. The authors highlight interesting important genetic determinants (rare and common) of variability in morphological traits, as well as experimentally validated several of rare variant associations using CRISPRi knockdown. The study is one of the first to examine genetic regulation of cellular morphological traits.

The reviewer greatly appreciates the effort put into the generation and analysis of this dataset and overall line of research. However, the manuscript was not carefully reviewed nor edited and there are a lot of necessary details missing. Moreover, there are a lot of statements made in the results and methods that are contradictory and/or not in alignment with the data shown in the figures and tables. Overall, this made the manuscript difficult to review. Given this, in my comments below I have not separated into major and minor issues because the manuscript needs a major revision. These are some of the issues but more existed with the manuscript.

A. Whole-genome sequencing and morphological profiling for 297 unique iPSC lines

Comments on Results

1. The authors presented "OnEdge" and "Residual" effects in Figure 2C and 2D. What are these variables and how were they determined? Consider describing these variables briefly somewhere.

Comments on Methods

1. Material and iPSC generation:

o The cohort has limited description. There should be a few sentences describing the male to female ratios, relatedness, ethnicities, and range of ages. This information is in tables and scattered throughout the results but should be summarized here in methods. Additionally, IRB approval and consent is not mentioned. There are a fair number of minors – were they broadly consented for this work.

o Which non-integrating episomal vectors were used? Is it Sendai virus? Please provide these details and/or appropriate references.

o Karyotyping results were not discussed or provided. What was the resolution for detecting rearrangements? And what are the results? It is expected that some lines will have large genome rearrangements (some of which are reoccurring), and these can significantly impact cell morphology.

o Y-27632: Please spell out that this is ROCK inhibitor or use the same nomenclature consistently throughout the manuscript.

o Since the goal is to analyze the morphology of the iPSCs the media and passaging schedules will affect how the cells and colonies appear. What was the media changing and passaging schedule?

2. Cell seeding and staining:

o What was the concentration of the ROCK Inhibitor used during plating cells on the 384-well plates?

o How many cells were seeded per well?

o If the cells were stained 6 hours after plating, how is it possible that they were in colonies after

such a short time? Typically, the cells would be barely attached. Moreover, the cells would easily detach during staining and washes steps of the Cell Painting procedure.

o Since ROCK inhibitor was used for the plating cells onto 384 well plates, the morphology of the cells would have been greatly affected by this treatment.

o Why were two nuclear DNA dyes used? Hoechst and SYTO 14 Green 436 Fluorescent Nucleic Acid Stain? Was SYTO 14 Green 436 Fluorescent Nucleic Acid Stain used to detect RNA?

o In line 426, it should read "cells were stained and imaged", not "cells were staining and imaged".

3. Cell Painting and imaging:

o You need to reference protocols that you refer to: "were imaged on a Perkin Elmer Phenix Automated Imaging system under a standardized protocol (REF)"

B. Principal components and variance component analyses

Comments on Results

1. The authors refer to the relationship that they identify between genetic variants and cellular morphology as "cell morphological quantitative trait loci (cmQTLs)". Technically for common variants the authors are conducting genome-wide association tests for 246 traits. These are not QTL studies, and it is confusing to refer them as such.

2. The meaning of this sentence is unclear: "We observed non-random segregation of iPSC lines in principal component analysis (PCA) of morphology traits across ancestry categories (Fig. S2) and across plates (Fig. S3), indicating the contribution of genetic and technical factors to the measurement of morphology." Fig. S2 has nothing to do with the morphology traits. BTW – Fig. S3 only show six of the seven plates. Additionally, the X-axis labels overlap each other. Consider decreasing the font or increasing the axis length.

3. Lines 122-123: "resulting in eight measurements per trait per donor, one for each of the eight replicates". This is the only location that mentions eight measurements per trait per donor were observed, i.e., it is not in the Methods, Supplemental Figures or anywhere. Considering that the plate accounts for the majority of the variability (61.8+17%) in all observed traits, were all replicates per donor on one plate or distributed across multiple plates?

Comments on Methods

1. Principal Component Analysis: Was the genetic relatedness matrix (GRM) used in the variance component and genetic association analyses.

2. Line 498: "The p-value of each factor was Bonferroni corrected for the number of traits." Is this for all 3,418 morphology traits or the 246 composite traits?

C. Rare variant association analysis

Comments on Results

1. It would be helpful to add a linear model equation for the rare variant analysis in the Methods. The authors state on line 159 that plate, well, and donor sex were controlled for. Does this mean that none of the other variables such as tissue of origin, ethnicity and relatedness were controlled for?

2. Line 160: "Of all tested traits, one trait in colony cells and 3 traits in isolated cells passed the genome-wide significance threshold ($P < 2.2 \times 10^{-8}$, Bonferroni correction for 246 traits and 9105 genes)". Figure 3A only shows 2 significant traits for isolated cells. Additionally, all four panels in Figure S7 are labeled "in isolate cells".

3. The isolated cells are 2.52% of all cells and defined as not being in contact with any other cells. Given that the majority of the genetic variance identified as associated with morphology traits was in these isolated cells, it is important to better understand the nature of these cells. A representative image of colony cells and isolated cells would be helpful. The concern is that single iPSC cells would typically be dead without the presence of ROCK inhibitor or spontaneously differentiated.

4. Lines 202 – 210: How was the size of RNA particles determined? This was a cytoplasm granularity category. The concern is that the dyes used to stain nucleic acids are not RNA specific. Hence, it is key to make sure that the phenotype being captured is not something else like mitochondrial DNA. A representative figure would be helpful.

5. Line 209: "Indeed, all iPSC lines carrying a rare variant burden in TSPAN15 had higher cell count compared to wild-type iPSC lines (Fig. S10)." Although the data look significant – they don't support this sentence.

6. Line 899: There are no orange dots in the figure.

7. In Figure S6, are arrows missing between 1) "cmQTL variants with MAF < 1%" and "Annotate variants with snpEff" and 2) "Filter non-synonymous variants with high and moderate effects" and "Filter non-synonymous variants with high effect"?

Comments on Methods

1. Lines 509-513 redundant. Same thing said twice.

D. Functional validation of rare variant associations

Comments on Results

1. Figures 4A and S8- are missing color legend for dyes. Figure S8 has inadequate figure legend, it is difficult to figure out the information that the authors want to convey.

2. Figures legends 4B-D – cannot figure out where these numbers are coming from "n = 56 non-targetsgRNAs, n = 56 targeting sgRNAs"

3. Lines 227-229: "We targeted each gene with 2 different sgRNAs, and validated each sgRNA for knockdown in expression of their gene target, showing a range of knockdown efficiency (15%-95%)". 15% of knock down at the RNA level is unlikely to alter the protein expression levels, and hence impact the phenotype.

4. Figure S12. It is not possible to understand the information being conveyed. For example, for WASF2 are you showing us two different gRNAs, and if yes, which housekeeping gene is used for the comparison? Or are you showing one gRNA compared to each of the two mentioned housekeeping genes? Why does TSPAN15 show different passages in the name? Are the authors trying to show that the gene expression goes from ~45% to about 75% within one passage? BTW- although GAPDH is a commonly used housekeeping control for qPCR, there is substantial literature showing that this is a highly variable genes whose expression is influenced by large number of factors, just for future reference.

Comments on Methods

1. Line 526: "(need to confirm these)" – the reviewer assumes that this has been confirmed.

2. sgRNA transduction in dCas9-iPSCs. Why did you switch to mTeSR1 for this method? Different culture conditions conduce different pluripotency states which would influence morphology. Also, please provide the company from which the mTeSR was procured.

E. Common variant association analysis

Comments on Results

1. It is a bit odd that no overlap was observed in the genetic variants associated with morphology in the iPSC cells that were in colonies versus those in isolated cells. See my comment above. Could this be due to spontaneous differentiation of the isolated cells?

Comments on Methods

1. It will be helpful to add a linear model equation including covariates for the common variant analysis in the Methods.

2. Line 502: "with 246 post-QC, INT traits that were described above" I assume that this is referring to lines 456-457: "Following this, each morphology trait was gaussianized across all 7 plates using inverse normal transformation (INT) method". I am not familiar with this method; would you please reference.

3. Line 504: How were the number of cell neighbors calculated? On lines 140-142 the authors indicated that there is one measurement per trait per donor. Within the same iPSC line, one would expect there to be huge variability in the number of cell neighbors?

F. Miscellaneous

1. Table legends insufficient. The Table headers are insufficient.

2. Table 2 should have donor ID provided – as in Table 1.

Reviewer #2:

Remarks to the Author:

Tegtmeyer et al describe the morphological analysis of iPSCs from 297 unique donor, and study how genetic variation impacts morphology. The authors describe 3 genes that contain rare-variants that affect morphology and report one common variant that does the same. Finally the authors suggest the presence of many more loci that affect cell morphology.

I am impressed by the size of this study, which represents a major step towards large-scale studies using iPSCs. Another novelty of the paper relies on the cell morphology analyses in conjunction with genetic analyses.

I do have a few questions and hope the answers to these questions will help to improve the paper. I believe these questions all have to do with the implications for future, even larger studies that are around the corner:

- The authors report three genes that contain rare variants and that affect certain morphological characteristics. The reported P-values are highly significant, yet when inspecting the provided box plots (thanks for providing those!), I am a tiny bit concerned that these effects might also potentially be caused by potentially non-normality of the data, and therefore some fluctuation of the non-parametric model that has been used to calculate significance. Probably my worry is unjustified, and I believe this can be shown fairly easily by the authors, by permuting the sample identifiers of the morphology data (i.e. breaking the link between the genotype and morphology data) and then redoing this burden analyses. What are the P-values that the authors then obtain by chance, are they sometimes equally, or nearly equally significant as the reported significances for WASF2, TSPAN15 or PRLR?

- The authors suggest the presence of many more common variants, because of over 300 loci attaining a $P < 4.1e-8$. However, given the numerous morphological phenotypes that have been studied, I do not really understand how this claim can be made that easily. Again it might help to again run permutations, and ascertain to what extent inflation of signal is observed (e.g. estimate

lambda inflation from QQ plot)? This might provide some formal basis for making this claim.

- Trans-eQTL and Trans-pQTLs are now successful, because sample sizes have become very large. Here the authors show that genetic variants can be identified that affect morphological phenotypes, using only a limited number of samples. I would very much appreciate it if the authors can speculate how this can be explained. Are these morphological phenotypes under stronger genetic control than gene expression levels? Or could it be that due to the use of iPSCs measurements are much more accurate? Or maybe less variable? It would be very valuable for the field to get some guidance on this, also with respect to the design of future experiments where ideally from iPSCs organoids can be developed at scale and a wide variety of single-cell multi-omics techniques, including cell painting could be applied. I am very much looking forward to that, and love to hear a bit more on what the authors believe would be adequate sample-sizes there.

Reviewer #3:

Remarks to the Author:

Review of Tegtmeyer et al High-dimensional phenotyping to define the genetic basis of cellular morphology for Nature Communications

The team assembled 297 iPSC lines from what looks like multiple sources<comments 1,2,3> to perform multidimensional analysis of cell morphology using established Cell Painting technology to link genetic variation to morphological phenotypes. They group also performed short read WGS on all of these lines. They identified 3 genes with rare variant burdens that correlated with a morphological phenotype<comment 4>. They validated each gene's effect on morphological phenotypes using CRISPRi knockdown experiments in a different iPSC line<comment 5>. They speculated on human phenotypes that may be associated with variants in these diseases but did not follow up by linking variation in these genes to human phenotypes by leveraging UKBB or AllofUs<comment 6,7>. The authors then associated common variants to morphological phenotypes and found 1, rs315506, at statistical significance. They speculate on the proposed mechanism <comment 8>. The authors state that many more variants almost reached statistical significance but would need a larger sample size <comment 9>

1. Please where the cells came from to the table that describes the cell lines. Especially since the cells labeled cmqtl... and BR00.... separate in PCA space.
2. Can the authors speculate on what the difference was between the the cmqtl... and BR00.... - derived cells? Were they induced differently? Stored differently?
3. Were there any measurements of common tissue culture parameters like growth rate, viability or freeze prior to going to Cell Painting? Were the cell lines similar? Would those measurements have been correlated to any of the measured morphology parameters? Do you think you could have weeded out some of the stochastic behavior?
4. Rare disease burden statistically associated morphological phenotypes in 2 genes and come pretty close in another. In all 3 cases they mention that several other linked traits had nominal P values for association. What is the distribution of P values for these genes versus genes that were not chosen for follow up? Were these genes special in having many nominally associated P values or not?
5. What does Cytoplasm_AreaShape_Zernicke_9_3 look like?
6. The authors do a lot of speculation about potential cellular and human disease phenotypes that may be associated with these genes and morphological phenotypes. A quick look at Genebase does not show traits associated to variation in WASF2, and alcoholism is the most associated trait to pLOF variants in PRLR. Is there a more sophisticated study one could do understand if the morphological changes associated with these genes are linked to disease phenotypes?
7. Were the authors surprised that the morphological traits did not correlate with any of the cell donor disease categories?
8. Regarding the descriptions of mechanisms for the phenotype associated with rs315506 in lines 255-257, it is really just speculation without any experimental follow up. Could you target regions of open chromatin in the 400KB window with CRISPRi/a to see if you can validate the phenotype?

Same for the other two noncoding variants mentioned.

9. Can the authors speculate on the sample size needed to reach significance for cm associations for another 10, 100 or 1000 common variants?

I think this is an important study. Understanding the functional effects of genetic variants is THE central challenge in human genetics. A ton of work went into the identification of these 3 statistically significant associations and the authors uncovered some key factors limiting their power. If I could ask for one addition it would be to add another section to the results on modeling how effect size, sample size and noise relate to power to detect cmQTLs to help guide future studies.

REVIEWER COMMENTS

Reviewer #1 (Remarks to the Author):

Tegtmeyer et al. conducted a cell morphological quantitative trait loci analysis investigating the associations in 297 iPSC lines from different donors between SNPs and 3,000 cellular morphological traits obtained using high-throughput imaging of multiple cellular compartments. The authors highlight interesting important genetic determinants (rare and common) of variability in morphological traits, as well as experimentally validated several of rare variant associations using CRISPRi knockdown. The study is one of the first to examine genetic regulation of cellular morphological traits.

The reviewer greatly appreciates the effort put into the generation and analysis of this dataset and overall line of research. However, the manuscript was not carefully reviewed nor edited and there are a lot of necessary details missing. Moreover, there are a lot of statements made in the results and methods that are contradictory and/or not in alignment with the data shown in the figures and tables. Overall, this made the manuscript difficult to review. Given this, in my comments below I have not separated into major and minor issues because the manuscript needs a major revision. These are some of the issues but more existed with the manuscript.

A. Whole-genome sequencing and morphological profiling for 297 unique iPSC lines

Comments on Results

1. The authors presented “OnEdge” and “Residual” effects in Figure 2C and 2D. What are these variables and how were they determined? Consider describing these variables briefly somewhere.

We apologize for the frustration experienced by the reviewer in understanding the paper. There are certainly a lot of details in this dense paper, and we appreciate being pointed to parts that were confusing. We had several authors read through the paper again to catch discrepancies and ambiguities, so the current version is much improved.

“OnEdge” refers to whether data came from a well which was located on either a row or column on the edge of the plate - this is a technical variable whose impact on the data we want to reduce. Residual is the remaining (technical) variance in morphological traits which is unexplained by the technical factors such as imaging batch, well position, and whether a sample was on the outer wells or demographic characteristics of our cell lines including genetic sex, ancestry, reprogramming sample source, age of donor at time of sample collection, and the clinical diagnosis for our tissue donors. We have updated the main text as well as the Figure 2 legend to clarify this.

We described these results in the main text on page 5:

“We assessed the significance for each variance component, correcting for the number of tests, which was the product of the traits ($n=3418$) and factors ($n=9$) which include technical features such as imaging plate, well position, the number of cell neighbors, and whether the well was positioned on the edge of the plate (onEdge) in addition to demographic characteristics for the cell lines including genetic sex, reprogramming sample source, age of donor at time of sample collection, and the clinical diagnosis for our tissue donors. We observed strong batch effects across imaging plates, which contributed the greatest degree of variance to our morphology traits ($61.8\pm 17\%$, **Figure 2B**, **Figure S3A**). Several other confounders contributed varying levels of effect on different morphological traits (**Figure 2C**). After correcting for these covariates, the remaining difference among cell line donors was significantly associated with all traits, explaining $16.7\pm 11\%$ of the variance. (**Figure 2B**). This indicated the potential for a genetic basis to the variability in morphology traits. Residual is the remaining (technical) variance in morphological traits which is unexplained by the factors discussed above.”

And in the Figure 2 legend on Page 16:

“**Figure 2. Summary of morphological traits and variant component analysis.** (A) Summary of five categories of morphological traits captured in our data ($n=3418$) (B) Exploring explained Variation in individual traits, namely distribution of mitochondria around nucleus, cytoplasmic Zernike shape metric 9_3, and cytoplasmic granularity in the RNA channel at scale 3, showed differences in sources of variance, including technical effects such as plate and well of imaging, whether the well was situated on the row or column on the edge of plate (onEdge), biological sources such as donor. Donor ID represents the difference among donors after accounting for their age, sex, disease-status and above-mentioned imaging-related technical factors. Residual is the remaining unaccounted variation in traits. (C) Same as C but for all morphological traits ($n=3418$).”

Comments on Methods

1. Material and iPSC generation:

1. The cohort has limited description. There should be a few sentences describing the male to female ratios, relatedness, ethnicities, and range of ages. This information is in tables and scattered throughout the results but should be summarized here in methods. Additionally, IRB approval and consent is not mentioned. There are a fair number of minors – were they broadly consented for this work.

We thank the reviewer for their comments. The cell lines used in our study do not require IRB approval as they have a non-human subjects research (NHRSR) designation.

We have updated the main text and the methods section to elaborate on the description of cell lines used in this study, including the distribution of sex, ancestry, and age across the cohort (page 4 and 21). This information is also captured in Table S1 and Figure S3.

All cell lines included in this work are available from, and were commercially purchased from, the California Institute for Regenerative Medicine (CIRM). We have also clarified this in the text. As per the consent forms maintained by CIRM, all donors included in this study have been properly consented for iPSC derivation, the experiments performed in this work, and genomic data sharing. We can confirm that all legal agreements are in place for the cell lines used in this study.

We described these results in the main text on page 4:

“We obtained pre-derived iPSCs from the CIRM iPSC repository. Age, sex, medical history, ethnicity, and relatedness to other samples were recorded using questionnaires at time of enrolment and sample collection. Each iPSC line undergoes a pluripotency test as well as genotyping to identify any abnormal karyotypes. Upon receipt, we expanded and cryo-banked each iPSC line, and performed genotyping (using the Global Screening Array (GSA)) and 30X whole-genome sequencing (WGS) on all lines. The final cohort used in this study included 297 distinct donors of which 153 were male and 144 were female, with an average age of 21 ± 10 (sd) years. Of the 297 donors, 207 had self-reported ancestry of European and 90 individuals reported non-European (**Table 1**). iPSC lines were generated from B-cells or fibroblasts using non-integrating episomal vector system as described in Lin et al, 2020 (**Table 1**). All donors included in this study have been properly consented for iPSC derivation, the experiments performed in this work, and genomic data sharing.”

2. Which non-integrating episomal vectors were used? Is it Sendai virus? Please provide these details and/or appropriate references.

To clarify this, we have emphasized in the text that these cell lines were purchased from the California Institute for Regenerative Medicine (CIRM) (Methods section, page 22), whose stem cell collection is described in detail here in Lin et al, 2020. Briefly, CIRM iPSCs were reprogrammed using FCDI's patented non-integrating episomal vector system described in Mack et al, 2011.

We described these results in the main text on page 4:

“iPSC lines were generated from B-cells or fibroblasts using non-integrating episomal vector system as described in Lin et al, 2020 (**Table 1**)”

Table 1. Summary of donors' sex, disease status, age and tissue used for iPSC generation. Total number of donors is 297

Donor metadata	Value
Sex	Male - 52% (n=153), Female - 48% (n=144)
Any disease	Yes - 62% (n=184), No - 38% (n=113)
Age	21±10
Self-reported ancestry	European - 70% (n=207), Non-European - 30% (n=90)
iPSC sample source	PBMCs- 62% (n=184), Fibroblasts- 38% (n=113)

3. Karyotyping results were not discussed or provided. What was the resolution for detecting rearrangements? And what are the results? It is expected that some lines will have large genome rearrangements (some of which are reoccurring), and these can significantly impact cell morphology.

We appreciate the reviewer for their comments and recognize the importance of discussing this. Each cell line included in our study was purchased from the CIRM iPSC repository where cells undergo extensive QC and karyotyping. Further, when cell lines are acquired in house, we apply our own rigorous QC which includes genotyping and whole-genome sequencing. Any cell lines which displayed abnormal karyotypes or harbored large genomic rearrangements (>1Mb) were excluded from our final cohort. While we cannot definitively rule out the possibility of new rearrangements acquired during the 3 days in culture leading up to plating in the 384 well format impacting morphology in our study, we have purposefully minimized the time in culture to capture the cells as close to the point of karyotyping (performed on at the time of cryopreservation) as possible (new Figure S2). Further, in the current study, our main focus was on germline genetic variation and exploring somatic mutations are out of the scope of this work. It would be great for future studies to explore somatic mutations and morphology in additional contexts.

All of the genotyping and whole-genome sequencing data for the cell lines incorporated into this study are publicly available here: https://app.terra.bio/#workspaces/broad-genomics-data/Eggan_iPSC_CIRM_GRU_GSA_MD.

We have updated the main text on page 3:

“Each iPSC line undergoes a pluripotency test as well as genotyping to identify any abnormal karyotypes. Upon receipt, we expanded and cryo-banked each iPSC line, and performed genotyping (using the Global Screening Array (GSA)) and 30X whole-genome sequencing (WGS) on all lines. Any cell lines displaying abnormal karyotypes or genomic rearrangements > 1Mb were excluded from our study. The final cohort used in

this study included 297 distinct donors of which 153 were male and 144 were female, with an average age of 21 ± 10 (sd) years.”

We have added this as a new limitation discussion in our main text on page 11:

“This study focused on how germline genetic variation influences stem cell morphology. While we applied a rigorous quality control to identify and remove cells with abnormal karyotypes or large genomic rearrangements, it is possible that new somatic mutations may arise overtime in culture. It will be important for additional studies to explore how recurring somatic mutations mediate cell morphology.”

We have updated our methods in the main text on page 20:

“Each iPSC sample underwent Global Screening Array (GSA) for karyotype analysis to ensure chromosomal integrity, as well as 30X whole-genome sequencing to determine genome-wide variants for each donor. Any cell lines displaying abnormal karyotypes or genomic rearrangements > 1Mb were excluded from our study. Each iPSC line was cultured between passages 12 and 15 before use in the experiment.”

4. Y-27632: Please spell out that this is ROCK inhibitor or use the same nomenclature consistently throughout the manuscript.

We have clarified these elements in the text for consistency.

We described these results in the main text on page 24:

“For each batch of imaging, cells were detached from 6-well NUNC plates using Accutase (StemcellTech; cat#07920) for generating single-cell suspensions. Following detachment, cells were centrifuged at 1000 rpm x 5:00 and re-suspended in StemFlex medium supplemented with 10uM ROCK inhibitor. After each cell line was counted to determine cell solution concentration and viability, the desired cell solution volume was aliquoted into a 96-deep well low attachment plate following a specific plate map to ensure that wells from any given cell line were not predominantly on the edge wells or too close together. To disperse a high number of cell lines across a 384-well plate in a semi-random fashion, we optimized the use of an Agilent Bravo liquid handling device (**Figure S1B**). Here, using an 8-channel head, cell solutions were transferred from the 96-well low attachment plate and distributed into a geltrex-coated Perkin Elmer Cell Carrier 384-well plate at a density of 10,000 cells per well. Each cell line was plated into 8 distinct wells on the final screening plate in four-well quadrants (see Figure S1B). These parameters were selected based on a pilot experiment with 6 cell lines across a range of densities and fixation conditions. We observed that we could maximize variability across cell lines using 10k cells per well fixed 6hrs after plating, when compared to 24hrs post-plating.”

5. Since the goal is to analyze the morphology of the iPSCs the media and passaging schedules will affect how the cells and colonies appear. What was the media changing and passaging schedule?

All cells included in our study were handled following the same workflows and passaging schedules - for example, iPSCs were thawed into a well of a 6 well plate and passaged 3 days later into a 96-well deep well plate before being plated onto a 384-well plate using an automated liquid handling device. The workflow was used for each of the 7 imaging batches for consistency. Cells were fed daily with StemFlex medium (or mTeSR1 medium for the dCas9-KRAB cell line). With our systematic workflow, cell lines were not passaged routinely prior to plating for the Cell Painting screen.

We described these results in the main text on page 5:

“Cell lines were thawed in batches of 48 and passaged 3 days later into a 96-well deep well plate before being transferred into a 384-well screening plate using an automated liquid handling device (**Figure S1B, Methods**).”

We describe additional details in the methods on page 24 (see Reviewer #1, Comment #4 for quoted text).

2. Cell seeding and staining:

1. What was the concentration of the ROCK Inhibitor used during plating cells on the 384-well plates?

The ROCK inhibitor concentration was the same during plating as the general culturing and passaging of the cells (10uM). This has been added to the methods within the text (page 23).

We described these results in the main text on page 23:

“Cells were cultured in StemFlex (ThermoFisher; cat#A334901) culture media and passaged for expansion with 1mM EDTA (Gibco; cat#15575020) and 10uM ROCK inhibitor (StemcellTech; cat#72304)..”

2. How many cells were seeded per well?

We thank the reviewer for this question. Cells were plated at an estimated density of 10,000 cells per well of the 384-well screening plate. We have added language within the methods section of the text (page 23) to clarify this.

We describe additional details in the methods on page 24 (see Reviewer #1, Comment #4 for quoted text):

3. If the cells were stained 6 hours after plating, how is it possible that they were in colonies after such a short time? Typically, the cells would be barely attached. Moreover, the cells would easily detach during staining and washes steps of the Cell Painting procedure.

We thank the reviewer for giving us the opportunity to clarify this point. While we passage the iPSCs as single cells, their migration and adherence to new plates occurs very quickly, often within two hours. Many cells will adhere near other cells and therefore represent "colonies" as defined in our current study (i.e., any cell which comes in contact with >2 other cells). There is very little detaching of cells while performing the staining and washing with a sufficient amount of care. We observed no patterns of cells washing off our wells during this process as can be observed from all images contained within our dataset, which can be found at <https://registry.opendata.aws/cellpainting-gallery/> under project ID "cpg0022-cmqtl." In our pilot screening, we tested various conditions related to the time of fixation post-plating, as well as the density. Our goal in this initial pilot was to identify the ideal time point which would maximize the variation across unique donor cell lines, enable us to quantify morphological features at both a single cell and colony level, and avoid proliferation/growth phenotypes. These observations are highlighted in the figure below (Figure 1). Using 6 cell lines, we tested two timepoints post-plating, 6hrs and 24hrs. In addition, we plated the cells across 8 different densities. One can see when examining the top two graphs that a 6hr time point was most poised to capture both isolated and colony cells (we determined that it would be valuable to capture data from both conformations) whereby at 24hrs the great majority of cells had already formed colonies. We have added language in the main text to refer to this pilot screen and more clearly articulate the parameters which were used during our discovery screen.

We have added clarifying language to the main text on page 4:

"It is important to clarify our definition of colony refers to the number of neighbors a given cell has and is not reflective of the idea of a colony which is often referred to in basic cell culture practices."

We described these results in the main text on page 5:

"We generated Cell Painting data from all 297 donors leveraging a systematic workflow to ensure cells were treated in identical fashion across all rounds of imaging. Cell lines were thawed in batches of 48 and passaged 3 days later into a 96-well deep well plate before being transferred into a 384-well screening plate using an automated liquid handling device (**Figure S1B, Methods**). Cells were plated at a density of 10k cells/per well and fixed 6 hrs post-plating, so as to allow for cell attachment while minimizing differences in cell growth or differentiation rates. We determined these conditions through a pilot screen that contained 6 cell lines plated across various densities and

fixation timepoints, which showed we could maximize differences between cell lines at these conditions (**Figure S1D**). Each screening plate was stained with the standard Cell Painting dyes and imaged on a Perkin Elmer Phenix automated microscope within 48hrs. We implemented this same workflow across all rounds of imaging. “

Figure 1 (text Figure S1D). UMAP plots for pilot screens to determine ideal time point and density. We tested several pilot conditions to identify the ideal parameters for our discovery cohort. We profiled cell lines derived from 6 donors and Cell Painted them under various densities (1000, 2000, 3000, 4000, 5000, 10000, 15000, and 20000 cells/well) and time points for fixation post-plating (6hr and 24hr). We leveraged this data to determine which conditions maximize our ability to measure cell line separation as well as reliably identify cells in both colony and isolate.

4. Since ROCK inhibitor was used for the plating of cells onto 384 well plates, the morphology of the cells would have been greatly affected by this treatment.

We appreciate the reviewers' comment. Indeed, ROCK inhibitors do affect the morphology of iPSCs. However, the primary aim of our work was to look for variation across many donors, who were all subjected to the same concentration of ROCK inhibitor for the same duration. Therefore, we believe that we are still in a strong position to compare across genetically diverse cell lines even though ROCK inhibitors may affect the morphology. We chose a condensed timeline between plating and fixing in an effort to avoid proliferation phenotypes whereby some cell lines would grow more quickly than

others and thus impact the morphology. As ROCK inhibitors are required for the survival of stem cells, we did not have an option to passage the cells without including it.

5. Why were two nuclear DNA dyes used? Hoechst and SYTO 14 Green 436 Fluorescent Nucleic Acid Stain? Was SYTO 14 Green 436 Fluorescent Nucleic Acid Stain used to detect RNA?

The reviewer is correct. Hoechst stains nuclear DNA and SYTO 14 is used to stain nucleoli and cytoplasmic RNA. SYTO14 has historically been used to measure intracellular RNA in a range of applications (see, Knowles et al, 1996, Tsuboi et al, 2015, and Knowles et al, 1997).

We have updated the text to clarify this on page 4:

“This multiplexing dye assay uses six stains to capture morphological characteristics for eight cellular compartments: Hoechst 33342 (DNA), wheat germ agglutinin (WGA) (golgi and plasma membrane), concanavalin A (endoplasmic reticulum), MitoTracker (mitochondria), SYTO 14 (nucleoli and cytoplasmic RNA), and phalloidin (actin).”

6. In line 426, it should read “cells were stained and imaged”, not “cells were staining and imaged”.

We thank the reviewer for pointing out this error. We have updated the text to address this.

3. Cell Painting and imaging:

1. You need to reference protocols that you refer to: “were imaged on a Perkin Elmer Phenix Automated Imaging system under a standardized protocol (REF)”

We thank the reviewer for catching this oversight. We have updated this line of text to share the appropriate reference.

We have updated the methods on page 21:

“Cell Painted plates were imaged on a Perkin Elmer Phenix Automated Microscope under a standardized protocol (Cimini et al, 2023).”

B. Principal components and variance component analyses

Comments on Results

1. The authors refer to the relationship that they identify between genetic variants and cellular morphology as “cell morphological quantitative trait loci (cmQTLs)”. Technically for common variants the authors are conducting genome-wide association tests for 246 traits. These are not QTL studies, and it is confusing to refer to them as such.

We thank the reviewer for their comments. Quantitative trait loci (QTL) studies apply to any genetic study of a quantitative trait, which uses a similar underlying methodology of regressing a phenotype on a genotype. There are many QTL studies which are not for expression QTLs (eQTLs) and are applied to genome-wide regressions against SNPs, such as protein QTLs or other quantitative traits such as height and blood traits (see, Yao et al, 2018, Yengo et al, 2022, Sinnott-Armstrong et al, 2022, Fairfax et al, 2014, Vosa et al, 2021). We believe it is valid to say we performed a QTL study, as we are identifying morphological traits which vary quantitatively with genetic variation.

2. The meaning of this sentence is unclear: “We observed non-random segregation of iPSC lines in principal component analysis (PCA) of morphology traits across ancestry categories (Fig. S2) and across plates (Fig. S3), indicating the contribution of genetic and technical factors to the measurement of morphology.” Fig. S2 has nothing to do with the morphology traits. BTW – Fig. S3 only show six of the seven plates. Additionally, the X-axis labels overlap each other. Consider decreasing the font or increasing the axis length.

We are grateful to the reviewer for their comments here and recognize some confusion due to the way this data was presented. Fig. S2 is not related to the morphological profiles but instead highlights the ancestry diversity for the 297 individuals within this study. It contains the first two PCs pertaining to the genetic data. Fig. S3 are the first two PCs pertaining to the morphological data highlighted by imaging batch to display plate to plate effects. We have updated this figure (now text Figure S3A) to include all 7 plates (see below). We have also clarified these elements within the text of the manuscript.

We described these results in the main text on page 6:

“We also observed strong batch effects across imaging plates (**Figure S3A**), indicating technical factors confound the measurement of morphology traits.”

Figure 2 (text Figure S3A). Principal component analysis of morphological traits across all 7 imaging batches. PCA showing batch effects in Cell Painting data across all 7 rounds of imaging. Each dot represents a cell line. This figure is an update to one contained in the initial manuscript but with additional annotations for clarity.

- Lines 122-123: “resulting in eight measurements per trait per donor, one for each of the eight replicates”. This is the only location that mentions eight measurements per trait per donor were observed, i.e., it is not in the Methods, Supplemental Figures or anywhere. Considering that the plate accounts for the majority of the variability (61.8+17%) in all observed traits, were all replicates per donor on one plate or distributed across multiple plates?

We thank the reviewer for their comments. The eight replicate measurements for trait per donor comes from each cell line being represented in 8 imaging wells. In each plate of imaging, we included 48 cell lines across 8 wells each. All replicates for a given donor are contained in the same plate and imaging run. We agree that distributing replicates of a given donor across plates would be ideal, but it’s a great technical challenge to keep 48 cell lines growing and to plate them at the same time, so we opted for the workflow described above which allowed us to maximize the number of cell lines included in any given imaging run while reducing the overall number of imaging runs. Please note that we included the plate as a covariate in our model, which allowed us to adjust for potential plate-to-plate variability, thereby ensuring the robustness of our results.

We described these methods more clearly in the main text on page 24:

“For each batch of imaging, cells were detached from 6-well NUNC plates using Accutase (StemcellTech; cat#07920) for generating single-cell suspensions. Following detachment, cells were centrifuged at 1000 rpm x 5:00 and re-suspended in StemFlex medium supplemented with 10uM ROCK inhibitor. After each cell line was counted to

determine cell solution concentration and viability, the desired cell solution volume was aliquoted into a 96-deep well low attachment plate following a specific plate map to ensure that wells from any given cell line were not predominantly on the edge wells or too close together. To disperse a high number of cell lines across a 384-well plate in a semi-random fashion, we optimized the use of an Agilent Bravo liquid handling device (**Figure S1B**). Here, using an 8-channel head, cell solutions were transferred from the 96-well low attachment plate and distributed into a geltrex-coated Perkin Elmer Cell Carrier 384-well plate at a density of 10,000 cells per well. Each cell line was plated into 8 distinct wells on the final screening plate in four-well quadrants (see Figure S2). These parameters were selected based on a pilot experiment with 6 cell lines across a range of densities and fixation conditions. We observed that we could maximize variability across cell lines using 10k cells per well fixed 6hrs after plating, when compared to 24hrs post-plating.”

We include a new supplemental figure on page 42:

Figure 3 (text Figure S1B). Cell culture workflow schematic. Overview of cell culture methods utilized in this study. Cell lines were thawed in batches of 48 and passaged 3 days later into 96-well deep well plates before being transferred to a 384-well screening plate using an automated liquid handling device (Agilent Bravo).

Comments on Methods

1. Principal Component Analysis: Was the genetic relatedness matrix (GRM) used in the variance component and genetic association analyses.

The GRM was implemented for the genetic PCA analysis but was not used in the variance component analysis nor the rare and common variant association test. We chose to exclude the GRM from the downstream analyses as much of the signal from the GRM would be captured in the first 4 genetics PCs which were included in these later analyses.

2. Line 498: “The p-value of each factor was Bonferroni corrected for the number of traits.” Is this for all 3,418 morphology traits or the 246 composite traits?

This was for all 3,418 morphological traits and we have clarified this in the text on page 20:

“The p-value of each factor was Bonferroni corrected for the number of all tested traits (n=3,418).”

C. Rare variant association analysis

Comments on Results

1. It would be helpful to add a linear model equation for the rare variant analysis in the Methods. The authors state on line 159 that plate, well, and donor sex were controlled for. Does this mean that none of the other variables such as tissue of origin, ethnicity and relatedness were controlled for?

We thank the reviewer for their comment. We have now included the linear model within the methods text (page 20) and here. Based on our initial results from our variance component analysis (text Figure 2), we only incorporated factors which contributed to trait variance in our dataset.

We described these results in the main text on page 20:

“Rare variant burden test

To perform the rare variant burden test, the variants which were autosomal, passed the VQSLOD filter and called in >95% individuals and had maf < 1% were retained. These variants were annotated for their functional effect using SnpEff v5.0. After annotation, those variants were kept which resided in the protein-coding region and had high or moderate effects on encoded protein. For each gene, multiple rare variants were grouped and coded as present or absent. The association between individual morphology traits and the presence of rare variants in a gene was investigated using linear regression models. The p-values of associations were corrected for both the number of tested traits (n = 246) and the number of genes (n = 9105) using Bonferroni correction method.”

Gaussianized trait ~ **Gene**_{variant present/absent} + **Age** + **Sex** + $\sum PC1-4$ + $(1/Plate)$ +

$(1/OnEdge_{proportion})$ + **Neighbor count**_{if not isolates}

2. Line 160: “Of all tested traits, one trait in colony cells and 3 traits in isolated cells passed the genome-wide significance threshold ($P < 2.2 \times 10^{-8}$, Bonferroni correction for 246 traits and 9105 genes)”. Figure 3A only shows 2 significant traits for isolated cells. Additionally, all four panels in Figure S7 are labeled “in isolate cells”.

We thank the reviewer for bringing this to our attention. The remaining dots were not named due to their association with the same gene, including 2 associations which overlap with one another for *PRLR*. For clarification, 3 traits associated with *PRLR*, and 1 trait had association with *WASF2*. We have better annotated Figure 3 to reflect this (pasted below). Additionally, we have corrected the supplemental figure which was wrongly labeled “isolate” cells for our colony association (also pasted below).

Figure 4 (text Figure 3). Updated figure for rare variant associations. A new version of the main Figure 3 in the paper. We have updated the annotations to provide more clarity on overlapping associations for rare variant burden in *PRLR*.

Figure 5 (text Figure S4A). QQ plots for rare variant associations. We have updated the supplemental figure (Figure S4A) based on the reviewer comment (the top left association was incorrectly labeled as isolate).

3. The isolated cells are 2.52% of all cells and defined as not being in contact with any other cells. Given that the majority of the genetic variance identified as associated with morphology traits was in these isolated cells, it is important to better understand the nature of these cells. A representative image of colony cells and isolated cells would be helpful. The concern is that single iPSC cells would typically be dead without the presence of ROCK inhibitor or spontaneously differentiated.

We thank the reviewer for their comment. We intentionally chose a 6hr time point post-fixation to ensure we could capture cells in either isolate or colony (>0 neighbors) format. Some of this was addressed in response to an earlier comment. Culturing the cells with ROCK inhibitor allowed the survival of isolate cells, especially at this short timepoint. Below, we have included an image (Figure 6 below) where we have annotated cells based on whether they are in isolate or colony to help clarify our classifications. We have also incorporated this into the text and as a new supplemental figure (**Figure S5**).

We described these results in the main text on page 5:

“To explore whether the contribution of genetic variation to cell morphology is context dependent, we segregated all cells into two groups based on whether they had any cells in contact (called colony cells, 97.48% of all cells) or not (called isolate cells, 2.52% of all cells) (**Figure S2B**).”

Figure 6 (text Figure S2B). Representative image of iPSCs in our dataset. This image is to highlight the variability in cell contexts which we categorize in our study. Cells which do not come in contact with other cells are classified as isolate cells. Those which touch 1 or more cells are classified as colony.

4. Lines 202 – 210: How was the size of RNA particles determined? This was a cytoplasm granularity category. The concern is that the dyes used to stain nucleic acids are not RNA specific. Hence, it is key to make sure that the phenotype being captured is not something else like mitochondrial DNA. A representative figure would be helpful.

We appreciate the reviewer's comment and are happy to add more clarity. We did not choose a single size of RNA particle; rather, we measured a whole size range in our feature extraction pipeline; the feature in question is one of 16 possible particle sizes. When examining the images for both the mitochondrial dyes and the cytoplasmic nucleic acid stain, we observed very little overlap (please see below), indicating this particular granularity metric seems unlikely to be driven by mitochondrial structures.

Figure 7. Display images for mitochondrial and cytoplasmic RNA dyes.

Representative images to highlight the lack of overlap in staining between the MitoTracker DeepRed mitochondrial and STYO14 nucleic acid stain.

- Line 209: “Indeed, all iPSC lines carrying a rare variant burden in TSPAN15 had higher cell count compared to wild-type iPSC lines (Fig. S10).” Although the data look significant – they don’t support this sentence.

We thank the reviewer for pointing out our mistake. We agree with this observation that our statement is not clearly supported by the data. We have now removed this from the text.

- Line 899: There are no orange dots in the figure.

Good catch. We have removed this statement from the legend of Figure S4E to reflect the data which is plotted.

- In Figure S6, are arrows missing between 1) “cmQTL variants with MAF < 1%” and “Annotate variants with snpEff” and 2) “Filter non-synonymous variants with high and moderate effects” and “Filter non-synonymous variants with high effect”?

We appreciate the reviewer bringing this to our attention.

We have corrected this in the schematic and integrated this into the methods on page 21:

Figure 8. Workflow for rare variant associations tests. Updated version of the supplementary figure displaying the workflow for testing morphological associations with rare variant burden. We have added arrows where they were missing thanks to the reviewer’s suggestion.

Comments on Methods

1. Lines 509-513 redundant. Same thing said twice.

We thank the reviewer for pointing out this grammatical error.

We have now fixed this within the text to remove redundancy on page 28:

“To perform the rare variant burden test, the variants which were autosomal, passed the VQSLOD filter and called in >95% individuals and had maf < 1% were retained. These variants were annotated for their functional effect using SnpEff v5.0. After annotation, those variants were kept which resided in the protein-coding region and had high or moderate effects on encoded protein.”

D. Functional validation of rare variant associations

Comments on Results

1. Figures 4A and S8- are missing color legend for dyes. Figure S8 has inadequate figure legend, it is difficult to figure out the information that the authors want to convey.

For clarity, we have altered Figure 4 to remove the Cell Painting images.

We have updated Figure S4C to contain color legends for each of the dyes on page 39:

Figure 9 (text Figure S4C). Representative images of cell lines carrying rare variants in *WASF2*, *PRLR*, and *TSPAN15*. We have added a color code to denote the various organelles and structures conveyed in the images. This figure was incorporated into the text to show that no striking differences in morphology could be observed by eye.

2. Figures legends 4B-D – cannot figure out where these numbers are coming from “n = 56 non-target sgRNAs, n = 56 targeting sgRNAs”

We thank the reviewer for their comments. The numbers in this figure refer to the replicate wells for each given perturbation. An “n=56” per target gene comes from including all wells for a given perturbation (n = 28 wells per targeting sgRNA -> 56 total wells per gene target).

We have updated the main text for clarity on page 10:

“For each of the three genes tested, we detected the predicted changes in each individual trait, and the change was in the same direction as our association analysis relative to non-targeting sgRNA controls (n = 28 wells per targeting sgRNA and 52 wells per non-targeting sgRNA, Welch’s Two Sample T-Test, $P < 2.2 \times 10^{-16}$) (**Fig. 4B-D**).”

And corrected the Figure 4 legend on page 15:

“Figure 4. Functional validation of rare-variant burden associations. (A) Workflow for knockdown of rare-variant genes using CRISPR interference in iPSCs expressing constitutive dCas9-KRAB. **(B-D)** Violin plots displaying quantification of traits between control non-targeting sgRNAs and sgRNAs targeting *WASF2*, *TSPAN15*, and *PRLR* on a per-well level ($n = 52$ wells per non-targeting sgRNAs, $n = 56$ wells per targeting sgRNAs, $P < 2.2 \times 10^{-16}$, Welch's Two-Sample T-Test). Effect on the trait score is consistent with what we observed in our rare-variant burden association. **(E)** Representative image of an observable gene-trait association for *PRLR*. Cells_RadialDistribution_RadialCV_Mito_1of4 relates to the asymmetric distribution of mitochondria in the ring right around the nucleus. In the non-targeting controls (left) we observed clustering of mitochondria on a particular side of the nucleus, whereas in the *PRLR* knockdown sgRNA (right) we observed a more distributed presence of mitochondria around the nucleus.”

3. Lines 227-229: “We targeted each gene with 2 different sgRNAs, and validated each sgRNA for knockdown in expression of their gene target, showing a range of knockdown efficiency (15%- 95%)”. 15% of knock down at the RNA level is unlikely to alter the protein expression levels, and hence impact the phenotype.

We thank the reviewer for their comment but respectfully disagree. Subtle changes in gene expression can account for a range of biological phenotypes. For example, in our own study, we observe quantitative changes ranging from 25-95% knockdown in our CRISPR interference validation experiment. Most heterozygous deletions elicit 50% knockdown in expression and can prove to have severe biological consequences. Gene expression is often stochastic and subtle changes in expression may lead to large changes in functional protein (see, Svenningsen et al, 2019, for example). Most expression QTL studies suggest very modest effects of a given variant on nearby gene expression (as low as 5-10%). It is, therefore, possible that a 15% reduction in expression may drive a biological phenotype.

We have updated our discussion in the main text on page 10:

“These associations were validated by CRISPR-mediated knockdown and supported by mechanistic information about these genes from the literature. Even though our effective knockdown had a range of efficiency (25-95%) we were able to measure meaningful changes in morphological features even at the lower range. This is consistent with previous work showing that gene expression is often stochastic and subtle changes in expression may lead to large changes in functional protein (Svenningsen et al, 2019).”

4. Figure S12. It is not possible to understand the information being conveyed. For example, for *WASF2* are you showing us two different gRNAs, and if yes, which

housekeeping gene is used for the comparison? Or are you showing one gRNA compared to each of the two mentioned housekeeping genes? Why does TSPAN15 show different passages in the name? Are the authors trying to show that the gene expression goes from ~45% to about 75% within one passage? BTW- although GAPDH is a commonly used housekeeping control for qPCR, there is substantial literature showing that this is a highly variable genes whose expression is influenced by large number of factors, just for future reference.

We appreciate the reviewers' feedback. For the RT-qPCR analyses, the Cq values for each sgRNA gene target was compared to the geometric mean for 2 different control genes; *GAPDH* and *RPL13* (we appreciate the heads up about GAPDH). The relative expression of each target gene was measured as described in Vandesompele et al, 2002 and Hellmens et al, 2007. The p27 and p28 in the *TSPAN15* name refers to the plasmid id and not the passage number.

We have updated Figure S5 in the text on page 39:

Figure 10 (text Figure S5). qPCR validation of CRISPR interference knockdown of rare-variant burden associations. We have updated this supplemental figure to correct labeling errors recognized by the reviewer. We have removed the extraneous identifiers for the *TSPAN15* sgRNAs (p27 and p28) which gave the impression of being related to passage number.

Comments on Methods

1. Line 526: “(need to confirm these)” – the reviewer assumes that this has been confirmed.

We thank the reviewer for bringing this to our attention and we have corrected the text to reflect the appropriate catalog numbers for the packaging plasmids.

2. sgRNA transduction in dCas9-iPSCs. Why did you switch to mTeSR1 for this method? Different culture conditions conduce different pluripotency states which would influence morphology. Also, please provide the company from which the mTeSR was procured.

The CRISPRi validation experiments were performed using mTeSR1 medium as our dCas9-KRAB (WTC11) cell line is recommended to be cultured using this medium (which was obtained by Stemcell Technologies, cat#: 85850; we have added this info on page 27). While it may alter morphology, all knockdown experiments were cultured in the same way such that we are observing relative changes in morphology. The validation of our rare variant associations using an alternative medium strengthens our result about genetic influences on cell morphology. It suggests that even under varying culture conditions, we can show that these variants mediate these phenotypes (and therefore the genetic influences may act independently of cell culture influences).

We have updated the text to clarify this on page 21:

“An iPSC line, WTC11_TO-NGN2_dCas9-BFP-KRAB (gift from Michael Ward), was seeded at 250k cells per well in a 12 well plate and 50ul of sgRNA lentivirus was added to each designated well. This iPSC line was cultured using mTeSR1 medium according to source recommendations (Stemcell Technologies, cat#: 85850).”

E. Common variant association analysis

Comments on Results

1. It is a bit odd that no overlap was observed in the genetic variants associated with morphology in the iPS cells that were in colonies versus those in isolated cells. See my comment above. Could this be due to spontaneous differentiation of the isolated cells?

We thank the reviewer for their comments. The increased detection of genetic associations with features in isolate is unlikely to be due to spontaneous differentiation. As the cells are fixed only 6 hrs after plating, there is likely not enough time to elicit any differentiation under these conditions. It may suggest that genetic factors which contribute to cell morphology may manifest themselves more clearly when cells are in isolate rather than in colonies. While the top feature associations (using a stringent Bonferroni correction) are quite different between the two cell contexts, there is a modest overall correlation between the associations of morphology traits and genetic variants (Figure 11 below).

We have updated the text on page 6:

“While the top feature associations (using a stringent Bonferroni correction) are quite different between isolate and colony cells, there is a modest overall correlation between the associations of morphology traits and genetic variants (**Figure S4B**).”

We display these results in Figure S4B on page 39:

B

Figure 11 (text Figure S4B). Correlation between all associations for rare variant burden in colony and isolate cells. This figure shows the overall correlation between the p values for each morphological trait and its association with a gene between cells in isolate and cells in colony. This shows there is modest overall correlation between both cell contexts.

Comments on Methods

1. It will be helpful to add a linear model equation including covariates for the common variant analysis in the Methods.

We have now added the linear model equations to the methods section page 20:

For isolated cells:

$$\text{Gaussianized trait} \sim \text{Variant} + \text{Age} + \text{Sex} + \sum \text{PC1-4} + (1|\text{Plate})$$

For cells in colony:

$$\text{Gaussianized trait} \sim \text{Variant} + \text{Age} + \text{Sex} + \sum \text{PC1-4} + (1|\text{Plate}) + \text{Neighbor count}$$

2. Line 502: “with 246 post-QC, INT traits that were described above” I assume that this is referring to lines 456-457: “Following this, each morphology trait was gaussianized

across all 7 plates using inverse normal transformation (INT) method". I am not familiar with this method; would you please reference.

We have provided details regarding this method below and now, in the methods. "The rank-based inverse normal transformation (INT) is commonly applied during GWAS of non-normally distributed traits. INT is a nonparametric mapping that replaces sample quantiles by quantiles from the standard normal distribution. After INT, the marginal distribution of any continuous outcome is asymptotically normal." We include key references also. The reference about INT is Operating characteristics of the rank-based inverse normal transformation for quantitative trait analysis in genome-wide association studies - McCaw - 2020 - Biometrics - Wiley Online Library. We used RankNorm function from RNOmni package from R. The detail of the function is RNOmni: Rank Normal Transformation Omnibus Test (r-project.org).

3. Line 504: How were the number of cell neighbors calculated? On lines 140-142 the authors indicated that there is one measurement per trait per donor. Within the same iPSC line, one would expect there to be huge variability in the number of cell neighbors?

We quantified cells' neighbors by measuring direct physical contact between cells in a field of imaging. For example, if a cell was in contact with 2 other cells, it would have 2 cell neighbors, etc. We extracted one measurement per trait per donor for each isolate and colony cells (meaning >0 cell neighbors identified). These cell-context feature measurements were then used for the association testing. We recognize that the ordering within the main text may confuse the reader.

We have updated the text to clarify this on page 5:

"To explore whether the contribution of genetic variation to cell morphology is context dependent, we segregated all cells into two groups based on whether they had any cells in contact (called colony cells, 97.48% of all cells) or not (called isolate cells, 2.52% of all cells) (**Figure S2B**). We note that for the purposes of our study, "colony" refers to the number of neighbors a given cell has and is distinct from the colony terminology which is often used in basic stem cell culture practices."

F. Miscellaneous

1. Table legends insufficient. The Table headers are insufficient.

These have been elaborated, as follows:

"Table S1. Metadata for cell lines incorporated in this study.

Metadata characteristics for donors from all 297 iPSC lines included in our study. Table

includes donor ID, sex, plate, stem cell source tissue, race, ethnicity, and clinical diagnosis.

Table S2. All morphological traits measured in our study.

List of all morphological traits measured in our imaging data which passed QC (n=3418).

Table S3. Composite traits for rare and common variant association tests.

Morphological traits which were selected for downstream association testing with rare and common variants. These features were chosen based on having a correlation of <0.9 with all other morphological traits contained in our dataset.

Table S4. Morphology traits with nominal association to rare variants in *WASF2*.

List of morphological traits which have nominal significance with association to rare variant burden in *WASF2*.

Table S5. Morphology traits with nominal association to rare variants in *PRLR*.

List of morphological traits which have nominal significance with association to rare variant burden in *PRLR*.

Table S6. Morphology traits with nominal association to rare variants.

List of all morphological traits which showed nominal association to rare variants in our study.

Table S7. Oligo sequences for sgRNAs used in CRISPRi experiments.

Sequences for sgRNAs used in the CRISPRi validation experiments.

Table S8. Morphology traits with suggestive evidence of association to common variants.

List of all morphological traits which show nominal association to common variants in our study. “

2. Table 2 should have donor ID provided – as in Table 1.

We did not have a Table 2 in the paper. Table S2 lists all of the morphological traits extracted from our imaging analysis and measured in our study (and not the morphological traits measured in each cell line). There is not any donor ID linked to this information. Perhaps the reviewer meant to highlight another provided table, in which we would be happy to add more information where appropriate.

Reviewer #2 (Remarks to the Author):

Tegtmeyer et al describe the morphological analysis of iPSCs from 297 unique donor, and study how genetic variation impacts morphology. The authors describe 3 genes that contain rare-variants that affect morphology and report one common variant that does the same. Finally the authors suggest the presence of many more loci that affect cell morphology.

I am impressed by the size of this study, which represents a major step towards large-scale studies using iPSCs. Another novelty of the paper relies on the cell morphology analyses in conjunction with genetic analyses.

I do have a few questions and hope the answers to these questions will help to improve the paper. I believe these questions all have to do with the implications for future, even larger studies that are around the corner:

1. The authors report three genes that contain rare variants and that affect certain morphological characteristics. The reported P-values are highly significant, yet when inspecting the provided box plots (thanks for providing those!), I am a tiny but concerned that these effects might also potentially be caused by potentially non-normality of the data, and therefore some fluctuation of the non-parametric model that has been used to calculate significance. Probably my worry is unjustified, and I believe this can be shown fairly easily by the authors, by permuting the sample identifiers of the morphology data (i.e. breaking the link between the genotype and morphology data) and then redoing this burden analyses. What are the P-values that the authors then obtain by chance, are they sometimes equally, or nearly equally significant as the reported significances for WASF2, TSPAN15 or PRLR?

We thank the reviewer for their comments. First, we want to reassure the reviewer that morphological traits were gaussianized before calculating their association with genetic variation. Hence non-normality is completely removed from the data before testing. Still, as suggested, we performed the permutation analysis by randomizing the values of morphology traits, which showed that significance of 4 observed associations is unlikely to occur by chance or as artifacts in a nonparametric model. We have now reported this analysis in text and appended the permutation plots to supplementary data (**Figure S12**, which is included below).

We described these results in the main text on page 7:

“To test whether the observed associations were an artifact of our nonparametric regression model, we permuted the data by randomly assigning trait values across donors. These results suggested that our observed significance of association between rare variant burden and cell morphological traits was unlikely to have occurred by chance (**Figure S4D**).”

And display the new results in Figure S4D on page 38:

D

Figure 12 (test Figure S4D). Distribution of permuted p values for rare variant associations. This figure displays the distribution of p values following permutations of our rare variant associations. The distribution of p values observed suggests that our associations are unlikely to have occurred by chance.

2. The authors suggest the presence of many more common variants, because of over 300 loci attaining a $P < 4.1e-8$. However, given the numerous morphological phenotypes that have been studied, I do not really understand how this claim can be made that easily. Again, it might help to again run permutations, and ascertain to what extent inflation of signal is observed (e.g. estimate lambda inflation from QQ plot)? This might provide some formal basis for making this claim.

We thank the reviewer for this astute comment. In line with the reviewers suggestion, we performed a permutation test by shuffling the genotypes and repeating our association analyses for both rare and common variants. First, we compared the mean distribution of lambda values between our true and permuted genotypes, which showed no pattern of overall inflation in our discovery set for both colony and isolate cells (**Figure 13**). Additionally, we used the lowest per-trait p values from our permutations to nominate a genome-wide 5% FDR threshold for our permuted associations. The genome-wide threshold from our permuted data in colony and isolate cells was $3.22e-6$ and $3.01e-6$, respectively. These are much higher than the genome-wide ($2e-10$) and suggestive ($4.1e-8$) thresholds measured in our previous analyses. These results indicate that our traits with suggestive evidence of association were not likely to have occurred by random chance.

We describe these results in the main text on page 9:

“To confirm our observed associations were not attributable to noise, we permuted the data by shuffling genotype labels and repeating the association tests. These results suggested that our observed significance of association between common variants and cell morphological traits was unlikely to have occurred by chance (Figure S7).”

We display these new results as a supplemental figure on page 42:

Figure 13 (text Figure S7). Permutation of common variant analyses. (A) Comparison of lambda value distributions between true and permuted genotypes in colony cells (mean lambda; true = 0.97, permuted = 0.99). **(B)** Comparison of lambda value distributions between true and permuted genotypes in isolate cells (mean lambda; true = 0.97, permuted = 1.01). **(C)** Distribution of the lowest p values and 5% FDR threshold based on our perturbation analysis (red = colony cells, blue = isolate cells). Genome-wide (black) and suggestive (orange) threshold from our discovery associations.

3. Trans-eQTL and Trans-pQTLs are now successful, because sample sizes have become very large. Here the authors show that genetic variants can be identified that affect morphological phenotypes, using only a limited number of samples. I would very much appreciate it if the authors can speculate how this can be explained. Are these morphological phenotypes under stronger genetic control than gene expression levels? Or could it be that due to the use of iPSCs measurements are much more accurate? Or maybe less variable? It would be very valuable for the field to get some guidance on this, also with respect to the design of future experiments where ideally from iPSCs organoids can be developed at scale and a wide variety of single-cell multi-omics techniques, including cell painting could be applied. I am very much looking forward to that, and love to hear a bit more on what the authors believe would be adequate sample-sizes there.

We thank the reviewer for sharing their thoughts. It is an intriguing and important question to address. While we are unsure if morphological traits are under stronger genetic control than other molecular features such as RNA or protein expression, we can

speculate a few things which may contribute to our discovery strength in this study, which we have expanded upon in our discussion. We show here that our discovery potential when leveraging whole-genome sequencing data is much greater than simply using SNP array data. This is due to our ability to map rare protein truncating variants, which are much more likely to elicit cellular phenotypes, to particular morphological traits. Many other QTL studies focus on common variants which often drive very subtle influences. Additionally, while our sample size is quite small relative to some of these studies, one benefit of a smaller sample size is to ensure that the wet lab techniques are systematically applied to all samples. We feel that our workflows are able to greatly reduce the noise introduced in cell culture systems, which enable us to be better able to identify associations here. There have been several recent studies exploring e/pQTLs in smaller sample sizes which have yielded promising results (see, Wolter et al, 2023, Mitchell et al, 2020, Wells et al, 2023, Jerber et al, 2021). We have added new results to our study to make quantitative predictions about our discovery potential for increasing sample sizes. A summary of these is posted below:

We described these results in the main text on page 9:

“Our findings suggest that genetic discovery for cell morphological phenotypes is achievable with a few hundred samples. However, the small number of significant discoveries in our analysis begs the question of how many discoveries can be made at larger sample sizes. If discovery of many hits requires a few thousand samples, such experiments are feasible and worthwhile; if such discovery requires hundreds of thousands to millions of samples, it may be out of reach for the foreseeable future. To answer this question, we estimated the distribution of common and rare variant effect sizes using Fourier Mixture Regression noLD (FMR-noLD) (**Methods; O’Connor, 2021**). Briefly, FMR-noLD fits a flexible mixture model to the distribution of effect sizes. This mixture model can be used to simulate effect sizes at various sample sizes, predicting how many significant discoveries will be made.

For common variants, we analyzed summary statistics from a pruned set of approximately 350,000 variants (**Methods**). We found that our dataset was underpowered for this analysis: FMR-noLD inferred that essentially all common variant effect sizes are 0, which is implausible and the expected behavior in the low power regime. For rare variants, we analyzed summary statistics from the main burden association analysis described earlier. In contrast to the common variant analysis, we predict that many discoveries will be made at feasible larger sample sizes, with more than 250 significant discoveries at N=1000 and more than 2000 discoveries at N=2000.”

We have updated our methods in the main text to describe these new results on page 22:

“Modeling cmQTL effect size distributions with FMR-noLD

We used FMR-noLD (O'Connor, 2021) to model the effect size distribution for both common and rare variant summary statistics from our analyses. FMR-noLD is a simplified version of the main FMR method that does not model linkage disequilibrium (LD) between variants. We used FMR-noLD rather than FMR for this analysis as 1) the mixed ancestry of our sample complicates LD-score style estimators such as FMR, and 2) FMR-noLD is the appropriate choice for rare variants, which have very little LD.

For the common-variant analysis, we used the PLINK2 (Chang et al, 2015) –indep-pairwise (with parameters: variant count 50, variant count shift 5, threshold 0.2) utility to find a set of approximately 350,000 variants in approximate linkage equilibrium. We then submitted the concatenated set of summary statistics across all traits for FMR-noLD. For the rare-variant analysis, we used the same set of summary statistics used in the main burden test analysis (i.e. with no need for variant pruning), concatenated across all traits.

For power analysis for rare variant association, we first predicted effect size distributions at varying sample sizes by adding sampling variance $1/N$ to our inferred distribution of true effect sizes. We then computed the cumulative distribution function of these predicted distributions at our significance threshold for the main rare variant burden analysis, $p = 2.2e-8$, which represents the proportion of tests that are expected to be significant at each sample size.”

We expanded on our discussion in the main text on page 11:

“These results suggest that genetic discovery for cell morphology is attainable at feasible sample sizes. Yet, the small number of significant discoveries in our study highlights that *in vitro* genetic studies still require substantial increases in sample sizes to gain discovery potential. Our common variant analysis suggests that we are vastly underpowered to measure genetic associations to cell morphology and our estimated effect size distributions infer that cell morphology may behave similarly to quantitative traits. As discovery potential for quantitative traits often scales linearly with the number of samples included in the study, our data suggests that even with 3000 genetically unique cell lines, we may still only yield 10 genome-wide significant common variant cmQTLs. This is a sobering result, as it suggests that tens of thousands of cell lines would be needed to begin mapping SNP-trait associations for cell morphology. In contrast, our analysis of rare variant effect sizes suggests that with modest increases in sample sizes, we are well-positioned to detect many rare variant cmQTLs. Future studies which can leverage 1000-2000 unique cell lines may yield many 1000s of genome-wide significant gene-trait associations. While scaling *in vitro* studies to 2000 cell lines will still be a large hurdle, it is one which can be feasibly overcome with current iPSC collections.”

Reviewer #3 (Remarks to the Author):

Review of Tegtmeyer et al High-dimensional phenotyping to define the genetic basis of cellular morphology for Nature Communications.

The team assembled 297 iPSC lines from what looks like multiple sources<comments 1,2,3> to perform multidimensional analysis of cell morphology using established Cell Painting technology to link genetic variation to morphological phenotypes. They group also performed short read WGS on all of these lines. They identified 3 genes with rare variant burdens that correlated with a morphological phenotype<comment 4>. They validated each gene's effect on morphological phenotypes using CRISPRi knockdown experiments in a different iPSC line<comment 5>. They speculated on human phenotypes that may be associated with variants in these diseases but did not follow up by linking variation in these genes to human phenotypes by leveraging UKBB or AllofUs<comment 6,7>. The authors then associated common variants to morphological phenotypes and found 1, rs315506, at statistical significance. They speculate on the proposed mechanism <comment 8>. The authors state that many more variants almost reached statistical significance but would need a larger sample size <comment 9>

1. Please where the cells came from to the table that describes the cell lines. Especially since the cells labeled cmqtl... and BR00.... separate in PCA space.

We thank the reviewer for their summary of the paper and this comment. The IDs were kept matching those associated with the publicly available image data, which do not reflect different cell line sourcing but rather a change in nomenclature during the course of the project to refer to imaging batches. We recognize this may be confusing and suggest the cell lines have varying origins. We have corrected this in the data tables to make it clearer. Additionally, we elaborated on the description of the cell lines used in our cohort in the text.

We described these results in the main text on page 3:

“We obtained pre-derived iPSCs from the CIRM iPSC repository. Age, sex, medical history, ethnicity, and relatedness to other samples were recorded using questionnaires at time of enrolment and sample collection. Each iPSC line undergoes a pluripotency test as well as genotyping to identify any abnormal karyotypes. Upon receipt, we expanded and cryo-banked each iPSC line, and performed genotyping (using the Global Screening Array (GSA)) and 30X whole-genome sequencing (WGS) on all lines. The final cohort used in this study included 297 distinct donors of which 153 were male and 144 were female, with an average age of 21 ± 10 (sd) years. Of the 297 donors, 207 had self-reported ancestry of European and 90 individuals reported non-European (**Table 1**). iPSC lines were generated from B-cells or fibroblasts using non-integrating episomal vector system as described in Lin et al, 2020 (**Table 1**). All donors included in this study have been properly consented for iPSC derivation, the experiments performed in this work, and genomic data sharing.”

2. Can the authors speculate on what the difference was between the cmqtl... and BR00...-derived cells? Were they induced differently? Stored differently?

See response to above comment. There is no difference in the cell lines, it was simply a workflow change in how plates were named by the imaging facility.

3. Were there any measurements of common tissue culture parameters like growth rate, viability or freeze prior to going to Cell Painting? Were the cell lines similar? Would those measurements have been correlated to any of the measured morphology parameters? Do you think you could have weeded out some of the stochastic behavior?

We appreciate the reviewer's suggestion. We have not recorded measurements for cell viability during cryopreservation or prior to plating for the Cell Painting screen. However, the number of cells plated for each experiment was calculated based on the live cell count at the time of the experiment. We do have growth rates calculated for each of the cell lines used in our study. We have included a histogram of the distribution below (also included as a new supplemental figure). These data show that doubling time ranges across cell lines and we used this to inform our experimental design. With this variability in mind, we fixed cells relatively quickly (6hrs) post-plating, in an effort to reduce proliferation or growth phenotypes which might confound our results. We believe the stochastic behavior with the most impact on cell morphology would be cell adherence and survival post-plating. Previous work in Vigilante et al, showed that interactions between cell adherence and the extracellular matrix could mediate cellular phenotypes. However, in our study at this scale it would have been a real challenge to accurately measure the ability for cells to adhere across hundreds of donors. Perhaps future study designs would include more robust measurements of other stem cell characteristics in order to measure their effects on cell morphology. For our purposes, whereby we wanted to identify relationships between genotype and cell morphology, we did not emphasize these types of analyses.

We display the distribution of growth rates in our cell lines in Figure S1C on page 36:

C

Figure 14 (text Figure S1C). Histogram of doubling time for cell lines used in this study. Distribution of cell lines based on growth rates calculated during time in culture before being screened for our study. These results show that doubling time does not typically vary more than 3-fold across lines in this study. We note that, even so, we optimized our cell culture workflows and plating timeline in an effort to prevent observing proliferation phenotypes in our workflows.

- Rare disease burden statistically associated morphological phenotypes in 2 genes and come pretty close in another. In all 3 cases they mention that several other linked traits had nominal P values for association. What is the distribution of P values for these genes versus genes that were not chosen for follow up? Were these genes special in having many nominally associated P values or not?

We appreciate the reviewer's inquiry about the distribution of P-values. Our analysis confirms a uniform distribution for genes not selected for follow-up, aligning with expectations for independent tests. This uniform distribution confirms that the observed associations are not an artifact of regression models. Furthermore, the genes WASF2 and PRLR showed a greater number of nominal associations ($p < 0.05$), indicating that rare variant burdens in these genes may influence various aspects of cellular morphology.

Figure 15. P-value distribution of morphological traits associated with rare variant burden. (Top) Distribution of P-values for all morphological traits in both colony and

isolate cells, which shows a uniform distribution. (Bottom) Distribution of pvals for associations with rare variants in *WASF2* and *PRLR*.

5. What does Cytoplasm_AreaShape_Zernicke_9_3 look like?

Zernike features are high dimensional mathematical descriptors, and as such, don't have easily-human-intuitable connections to cell shapes. Below is an example of reconstructing an image of a horse (first image) with its regular Zernike_9_3, after inverting Zernike_9_3, after multiplying Zernike_9_3 by 10, and after dividing it by 10. More detail is available at this link. CellProfiler Zernikes are reported in a slightly different manner than used here (CellProfiler Zernikes are not mathematically easily reconstructable), but the principle holds.

Figure 16. Graphical depiction of Zernike features. This figure helps to describe how Zernike features are represented in organelle shape and size. While difficult to discern by eye, Zernike features represent polynomial reconstructions of images.

6. The authors do a lot of speculation about potential cellular and human disease phenotypes that may be associated with these genes and morphological phenotypes. A quick look at Genebase does not show traits associated to variation in *WASF2*, and alcoholism is the most associated trait to pLOF variants in *PRLR*. Is there a more sophisticated study one could do understand if the morphological changes associated with these genes are linked to disease phenotypes?

We acknowledge the reviewer's suggestion that a more sophisticated analysis linking morphological changes to specific disease phenotypes would be valuable. While our study does not undertake this in-depth investigation, we believe our findings lay the groundwork for further research. Specifically, our identification of rare-variant burdens in *WASF2*, *PRLR*, and *TSPAN15* and their potential impact on cellular morphology provides new insights into genes that may be associated with various cancers and tumor-related disorders.

WASF2 is named for its relationship to Wiskott-Aldrich syndrome, which increases the risk of various cancers. The novel finding that rare variants in this gene impact actin cytoskeleton and cell area supports its role in cancer-related conditions. Likewise, *PRLR* is associated with breast cancer and lymphoma, and our result showing that loss of *PRLR* impacts mitochondrial distribution throughout the cell supports previous work

suggesting altered mitochondrial activity in breast cancer progression. TSPAN15, a member of the tetraspanin family, has been implicated in tumor-related conditions.

We hope our findings serve as a foundation for experts to better understand the cellular mechanisms of these disorders. Our results may also suggest that genetic studies in stem cell models can shed insights into cancer-linked genes, and future studies could explore these associations in different cell types or disease-relevant tissues. We have updated our discussion in the main text to reflect these insights and clarify the current scope and potential future directions of our research.

We have updated our results in the main text on page 6:

“*WASF2* is named for its association to Wiskott-Aldrich syndrome, a rare genetic disorder which greatly increases the risk of various cancers (Ding et al, 2023, Yang et al, 2022, Rana et al, 2021, and Rana et al, 2023).”

“*PRLR* function has been linked to several forms of cancers, including breast cancer and lymphoma (Kavarthapu et al, 2022, López Fontana et al, 2021, Gharbaran et al, 2021).”

We have updated our results in the main text on page 6:

“A member of the tetraspanin family of transmembrane segments, TSPAN15 has been implicated in tumor related conditions (Huang et al, 2022).”

We have updated our discussion in the main text on page 7:

“Each of the genes implicated in our rare variant analysis have been implicated in various cancers. We find this result interesting and somewhat unsurprising, given that pluripotent stem cells exhibit self-renewal properties which closely resemble cancer cells. These results suggest that genetic studies in iPSCs may shed meaningful insights into cancer-linked genes. It will be important for future studies to measure whether these associations are cell-type specific, or if they would be retained using differentiated cells.”

7. Were the authors surprised that the morphological traits did not correlate with any of the cell donor disease categories?

We are likely underpowered in this study to detect any disease associations. The 297 samples have many [6] different clinical disease categories represented in this cohort, meaning that we have too few cell lines in any given category. Furthermore, there have been a few studies that have identified morphologies associated with disease (please see, Gharaba et al, 2023, Antony et al, 2020, Schiff, Migliori, Chen, Carter et al, 2022), but it's not clear whether every disorder will have a morphology detectable by Cell Painting, especially without differentiating into more specialized, disease-relevant cell

types or having increased samples sizes per clinical disease category. Overall, this is a great point of discussion and we have expanded on this to highlight these critical next steps for implementing this assay for disease modeling.

We have expanded on our discussion in the main text on page 11:

“Moreover, we did not find any cell morphological traits associated with clinical disease categories from the cell line donors (data not shown). Similar to exploring common variant associations, we are likely underpowered in any single disease category to identify significant associations. There have been many studies elucidating morphological features associated with various diseases, but they often contained larger sample sizes and incorporated more specialized cell types (Gharaba et al, 2023, Antony et al, 2020, Schiff, Migliori, Chen, Carter et al, 2022). Extending our current study to diverse cell types and increasing the number of samples for clinical disease categories will be a critical next step in efforts to link cell morphology to human illnesses.”

8. Regarding the descriptions of mechanisms for the phenotype associated with rs315506 in lines 255-257, it is really just speculation without any experimental follow up. Could you target regions of open chromatin in the 400KB window with CRISPRi/a to see if you can validate the phenotype? Same for the other two noncoding variants mentioned.

We thank the reviewer for their comments here. We recognize that our interpretation of our common variant association is speculative. However, using CRISPRi/a to perturb regions of open chromatin around rs315506 would be a very time and resource intensive experiment which is outside the scope for this study. However, to explore linking our common variant association more closely with a biological phenotype, we queried publicly available Cell Painting data on U2OS cells perturbed using CRISPR interference. This dataset included two genes that lie within 400kb of our significant SNP-trait association (*NF1* and *SUZ12*). Additionally, it included a gene which contains a SNP with suggestive evidence for association with *Nuclei_Granularity_9_AGP*. Upon knockdown of *NF1* and *SUZ12*, we observed a significant change in the morphological trait nominated in our morphology association with the nearby SNP rs315506 (*Cytoplasm_RadialDistribution_RadialCV_ER_3of4*). Both knockdowns impact this morphological trait in the same direction. These results provide further experimental support to the possibility that differences in gene expression at this locus, as a result of either genetic perturbation, or common variation may alter the radial distribution of the endoplasmic reticulum. We did not observe a statistically significant change in feature score for *Nuclei_Granularity_9_AGP* when there was a knockdown of *PRKAR1B*.

We described these results in the main text on page 8:

“To corroborate this observation, we analyzed the publicly available JUMP-Cell Painting data from U2OS cells that have perturbed *NF1* and *SUZ12* using CRISPR interference

(Chandrasekaran et al., 2023). In this data, we see a significant change in our associated trait when NF1 and SUZ12 expression is decreased (**Figure 5B**).”

And

“We were unable to link perturbations in *PRKAR1B* to morphological changes for this feature using publicly available data (**Figure S6B**).”

Figure 17 (text Figure 5B). Knockdown of genes nearby common variant associations. Impact of knockdown of *NF1* and *SUZ12* in U2OS cells on *Cytoplasm_RadialDistribution_RadialCV_ER_3of4* ($P = 0.04$ *NF1*, $P = 0.005$ *SUZ12*, Welch's Two-Sample T-Test). Data is presented in a Tukey-style boxplot with the median (Q2) and the first and the second quartiles (Q1, Q3) and error bars defined by the last data point within ± 1.5 -times the interquartile range.

Figure 18 (text Figure S6B) (right). Knockdown of genes nearby common variant associations. Impact of knockdown of *PRKAR1B* in U2OS cells on *Nuclei_Granularity_9_AGP* ($P=0.60$, Welch's Two-Sample T-Test). Data is presented in a Tukey-style boxplot with the median (Q2) and the first and the second quartiles (Q2, Q3) and error bars defined by the last data point within ± 1.5 -times the interquartile range.

9. Can the authors speculate on the sample size needed to reach significance for cm associations for another 10, 100 or 1000 common variants?

We thank the reviewer for their comments and share in their interest to better understand sample size requirements for these types of studies. We decided to invest time in making appropriate and quantitatively grounded estimates for this.

We described these results in the main text on page 9:

“Our findings suggest that genetic discovery for cell morphological phenotypes is achievable with a few hundred samples. However, the small number of significant discoveries in our analysis begs the question of how many discoveries can be made at larger sample sizes. If discovery of many hits requires a few thousand samples, such experiments are feasible and worthwhile; if such discovery requires hundreds of thousands to millions of samples, it may be out of reach for the foreseeable future. To answer this question, we estimated the distribution of common and rare variant effect sizes using Fourier Mixture Regression noLD (FMR-noLD) (**Methods; O'Connor, 2021**). Briefly, FMR-noLD fits a flexible mixture model to the distribution of effect sizes. This mixture model can be used to simulate effect sizes at various sample sizes, predicting how many significant discoveries will be made.

For common variants, we analyzed summary statistics from a pruned set of approximately 350,000 variants (**Methods**). We found that our dataset was underpowered for this analysis: FMR-noLD inferred that essentially all common variant effect sizes are 0, which is implausible and the expected behavior in the low power regime. For rare variants, we analyzed summary statistics from the main burden association analysis described earlier. In contrast to the common variant analysis, we predict that many discoveries will be made at feasible larger sample sizes, with more than 250 significant discoveries at N=1000 and more than 2000 discoveries at N=2000.”

Figure 18. Distribution of effect sizes and predictions for future discovery. A) QQ plot with current observed tests, model fit, and three predicted lines corresponding to N = 500, 1000, 2000. **B)** Boxplots showing the number of discoveries per trait at current N, then projected N = 500, 1000, 2000.

We expanded on our discussion in the main text on page 11:

“These results suggest that genetic discovery for cell morphology is attainable at feasible sample sizes. Yet, the small number of significant discoveries in our study highlights that *in vitro* genetic studies still require substantial increases in sample sizes to gain discovery potential. Our common variant analysis suggests that we are vastly underpowered to measure genetic associations to cell morphology and our estimated effect size distributions infer that cell morphology may behave similarly to quantitative traits. As discovery potential for quantitative traits often scales linearly with the number of samples included in the study, our data suggests that even with 3000 genetically unique cell lines, we may still only yield 10 genome-wide significant common variant cmQTLs. This is a sobering result, as it suggests that tens of thousands of cell lines would be needed to begin mapping SNP-trait associations for cell morphology. In contrast, our analysis of rare variant effect sizes suggests that with modest increases in

sample sizes, we are well-positioned to detect many rare variant cmQTLs. Future studies which can leverage 1000-2000 unique cell lines may yield many 1000s of genome-wide significant gene-trait associations. While scaling in vitro studies to 2000 cell lines will still be a large hurdle, it is one which can be feasibly overcome with current iPSC collections.”

We have updated our methods in the main text to describe these new results on page 22:

“Modeling cmQTL effect size distributions with FMR-noLD

We used FMR-noLD (O’Connor, 2021) to model the effect size distribution for both common and rare variant summary statistics from our analyses. FMR-noLD is a simplified version of the main FMR method that does not model linkage disequilibrium (LD) between variants. We used FMR-noLD rather than FMR for this analysis as 1) the mixed ancestry of our sample complicates LD-score style estimators such as FMR, and 2) FMR-noLD is the appropriate choice for rare variants, which have very little LD.

For the common-variant analysis, we used the PLINK2 (Chang et al, 2015) –indep-pairwise (with parameters: variant count 50, variant count shift 5, threshold 0.2) utility to find a set of approximately 350,000 variants in approximate linkage equilibrium. We then submitted the concatenated set of summary statistics across all traits for FMR-noLD. For the rare-variant analysis, we used the same set of summary statistics used in the main burden test analysis (i.e. with no need for variant pruning), concatenated across all traits.

For power analysis for rare variant association, we first predicted effect size distributions at varying sample sizes by adding sampling variance $1/N$ to our inferred distribution of true effect sizes. We then computed the cumulative distribution function of these predicted distributions at our significance threshold for the main rare variant burden analysis, $p = 2.2e-8$, which represents the proportion of tests that are expected to be significant at each sample size.”

I think this is an important study. Understanding the functional effects of genetic variants is THE central challenge in human genetics. A ton of work went into the identification of these 3 statistically significant associations and the authors uncovered some key factors limiting their power. If I could ask for one addition it would be to add another section to the results on modeling how effect size, sample size and noise relate to power to detect cmQTLs to help guide future studies.

Lea Starita, University of Washington

References:

Lin SS, DeLaura S, Jones EM. The CIRM iPSC repository. *Stem Cell Res.* 2020 Apr;44:101671. doi: 10.1016/j.scr.2019.101671. Epub 2019 Nov 26. PMID: 32151950.

Mack AA, Kroboth S, Rajesh D, Wang WB. Generation of induced pluripotent stem cells from CD34+ cells across blood drawn from multiple donors with non-integrating episomal vectors. *PLoS One.* 2011;6(11):e27956. doi: 10.1371/journal.pone.0027956. Epub 2011 Nov 22. PMID: 22132178; PMCID: PMC3222670.

Svenningsen MS, Semsey S, Mitarai N. Gene Expression Changes with Minor Effects on the Population Average Have Major Effects on the Occurrence of Cells with Extreme Protein Concentrations. *mSphere.* 2019 Jan 30;4(1):e00575-18. doi: 10.1128/mSphere.00575-18. PMID: 30700510; PMCID: PMC6354807.

Vandesompele J, De Preter K, Pattyn F, Poppe B, Van Roy N, De Paepe A, Speleman F. Accurate normalization of real-time quantitative RT-PCR data by geometric averaging of multiple internal control genes. *Genome Biol.* 2002 Jun 18;3(7):RESEARCH0034. doi: 10.1186/gb-2002-3-7-research0034. Epub 2002 Jun 18. PMID: 12184808; PMCID: PMC126239.

Hellemans J, Mortier G, De Paepe A, Speleman F, Vandesompele J. qBase relative quantification framework and software for management and automated analysis of real-time quantitative PCR data. *Genome Biol.* 2007;8(2):R19. doi: 10.1186/gb-2007-8-2-r19. PMID: 17291332; PMCID: PMC1852402.

Fairfax BP, Humburg P, Makino S, Naranbhai V, Wong D, Lau E, Jostins L, Plant K, Andrews R, McGee C, Knight JC. Innate immune activity conditions the effect of regulatory variants upon monocyte gene expression. *Science.* 2014 Mar 7;343(6175):1246949. doi: 10.1126/science.1246949. PMID: 24604202; PMCID: PMC4064786.

Võsa U, Claringbould A, Westra HJ, Bonder MJ, Deelen P, Zeng B, Kirsten H, Saha A, Kreuzhuber R, Yazar S, Brugge H, Oelen R, de Vries DH, van der Wijst MGP, Kasela S, Pervjakova N, Alves I, Favé MJ, Agbessi M, Christiansen MW, Jansen R, Seppälä I, Tong L, Teumer A, Schramm K, Hemani G, Verluuw J, Yaghootkar H, Sönmez Flitman R, Brown A, Kukushkina V, Kalnapienik A, Rieger S, Porcu E, Kronberg J, Kettunen J, Lee B, Zhang F, Qi T, Hernandez JA, Arindrarto W, Beutner F; BIOS Consortium; i2QTL Consortium; Dmitrieva J, Elansary M, Fairfax BP, Georges M, Heijmans BT, Hewitt AW, Kähönen M, Kim Y, Knight JC, Kovacs P, Krohn K, Li S, Loeffler M, Marigorta UM, Mei H, Momozawa Y, Müller-Nurasyid M, Nauck M, Nivard MG, Penninx BWJH, Pritchard JK, Raitakari OT, Rotzschke O, Slagboom EP, Stehouwer CDA, Stumvoll M, Sullivan P, 't Hoen PAC, Thiery J, Tönjes A, van Dongen J, van Iterson M, Veldink JH, Völker U, Warmerdam R, Wijmenga C, Swertz M, Andiappan A, Montgomery GW, Ripatti S, Perola M, Kutalik Z, Dermizakis E, Bergmann S, Frayling T, van Meurs J, Prokisch H, Ahsan H, Pierce BL, Lehtimäki T, Boomsma DI, Psaty BM, Gharib SA, Awadalla P, Milani L, Ouwehand WH, Downes K, Stegle O, Battle A, Visscher PM, Yang J, Scholz M, Powell J, Gibson G, Esko T, Franke L. Large-scale cis- and trans-eQTL analyses identify thousands of genetic loci and polygenic scores that regulate blood gene expression. *Nat Genet.*

2021 Sep;53(9):1300-1310. doi: 10.1038/s41588-021-00913-z. Epub 2021 Sep 2. PMID: 34475573; PMCID: PMC8432599.

Ding G, Wang T, Liu S, Zhou Z, Ma J, Wu J. Wiskott-Aldrich syndrome gene as a prognostic biomarker correlated with immune infiltrates in clear cell renal cell carcinoma. *Front Immunol*. 2023 Apr 12;14:1102824. doi: 10.3389/fimmu.2023.1102824. PMID: 37122750; PMCID: PMC10130519.

Yang X, Ding Y, Sun L, Shi M, Zhang P, He A, Zhang X, Huang Z, Li R. WASF2 Serves as a Potential Biomarker and Therapeutic Target in Ovarian Cancer: A Pan-Cancer Analysis. *Front Oncol*. 2022 Mar 14;12:840038. doi: 10.3389/fonc.2022.840038. PMID: 35359421; PMCID: PMC8964075.

Rana PS, Alkrekshi A, Wang W, Markovic V, Sossey-Alaoui K. The Role of WAVE2 Signaling in Cancer. *Biomedicines*. 2021 Sep 14;9(9):1217. doi: 10.3390/biomedicines9091217. PMID: 34572403; PMCID: PMC8464821.

Rana PS, Wang W, Markovic V, Szpendyk J, Chan ER, Sossey-Alaoui K. The WAVE2/miR-29/Integrin- β 1 Oncogenic Signaling Axis Promotes Tumor Growth and Metastasis in Triple-negative Breast Cancer. *Cancer Res Commun*. 2023 Jan 31;3(1):160-174. doi: 10.1158/2767-9764.CRC-22-0249. PMID: 36968231; PMCID: PMC10035451.

Gharaba S, Paz O, Feld L, Abashidze A, Weinrab M, Muchtar N, Baransi A, Shalem A, Sprecher U, Wolf L, Wolfenson H, Weil M. Perturbed actin cap as a new personalized biomarker in primary fibroblasts of Huntington's disease patients. *Front Cell Dev Biol*. 2023 Jan 18;11:1013721. doi: 10.3389/fcell.2023.1013721. PMID: 36743412; PMCID: PMC9889876.

Antony PMA, Kondratyeva O, Mommaerts K, Ostaszewski M, Sokolowska K, Baumuratov AS, Longhino L, Poulain JF, Grossmann D, Balling R, Krüger R, Diederich NJ. Fibroblast mitochondria in idiopathic Parkinson's disease display morphological changes and enhanced resistance to depolarization. *Sci Rep*. 2020 Jan 31;10(1):1569. doi: 10.1038/s41598-020-58505-6. PMID: 32005875; PMCID: PMC6994699.

Schiff L, Migliori B, Chen Y, Carter D, Bonilla C, Hall J, Fan M, Tam E, Ahadi S, Fischbacher B, Geraschenko A, Hunter CJ, Venugopalan S, DesMarteau S, Narayanaswamy A, Jacob S, Armstrong Z, Ferrarotto P, Williams B, Buckley-Herd G, Hazard J, Goldberg J, Coram M, Otto R, Baltz EA, Andres-Martin L, Pritchard O, Duren-Lubanski A, Daigavane A, Reggio K; NYSCF Global Stem Cell Array® Team; Nelson PC, Frumkin M, Solomon SL, Bauer L, Aiyar RS, Schwarzbach E, Noggle SA, Monsma FJ Jr, Paull D, Berndl M, Yang SJ, Johannesson B. Integrating deep learning and unbiased automated high-content screening to identify complex disease signatures in human fibroblasts. *Nat Commun*. 2022 Mar 25;13(1):1590. doi: 10.1038/s41467-022-28423-4. PMID: 35338121; PMCID: PMC8956598.

Knowles RB, Sabry JH, Martone ME, Deerinck TJ, Ellisman MH, Bassell GJ, Kosik KS. Translocation of RNA granules in living neurons. *J Neurosci*. 1996 Dec 15;16(24):7812-20. doi: 10.1523/JNEUROSCI.16-24-07812.1996. PMID: 8987809; PMCID: PMC6579227.

Tsuboi D, Kuroda K, Tanaka M, Namba T, Iizuka Y, Taya S, Shinoda T, Hikita T, Muraoka S, Iizuka M, Nimura A, Mizoguchi A, Shiina N, Sokabe M, Okano H, Mikoshiba K, Kaibuchi K. Disrupted-in-schizophrenia 1 regulates transport of ITPR1 mRNA for synaptic plasticity. *Nat Neurosci*. 2015 May;18(5):698-707. doi: 10.1038/nn.3984. Epub 2015 Mar 30. PMID: 25821909.

Knowles RB, Kosik KS. Neurotrophin-3 signals redistribute RNA in neurons. *Proc Natl Acad Sci U S A*. 1997 Dec 23;94(26):14804-8. doi: 10.1073/pnas.94.26.14804. PMID: 9405694; PMCID: PMC25118.

Xing J, Qi L, Liu X, Shi G, Sun X, Yang Y. Roles of mitochondrial fusion and fission in breast cancer progression: a systematic review. *World J Surg Oncol*. 2022 Oct 3;20(1):331. doi: 10.1186/s12957-022-02799-5. PMID: 36192752; PMCID: PMC9528125.

Kavarthapu R, Dufau ML. Prolactin receptor gene transcriptional control, regulatory modalities relevant to breast cancer resistance and invasiveness. *Front Endocrinol (Lausanne)*. 2022 Sep 15;13:949396. doi: 10.3389/fendo.2022.949396. PMID: 36187116; PMCID: PMC9520000.

López Fontana G, Rey L, Santiano F, López Fontana R, López Laur JD, Zyla L, Valdemoros P, Guerrero-Giménez ME, Fernández-Muñoz JM, Gómez S, Bruna F, Guglielmi JM, Carón R, López Fontana C. Changes in prolactin receptor location in prostate tumors. *Arch Esp Urol*. 2021 May;74(4):419-426. English, Spanish. PMID: 33942735.

Gharbaran R, Onwumere O, Codrington N, Somenarain L, Redenti S. Immunohistochemical localization of prolactin receptor (PRLR) to Hodgkin's and Reed-Sternberg cells of Hodgkin's lymphoma. *Acta Histochem*. 2021 Jan;123(1):151657. doi: 10.1016/j.acthis.2020.151657. Epub 2020 Nov 28. PMID: 33259941.

Reviewers' Comments:

Reviewer #1:

Remarks to the Author:

The authors have done a thorough job with the edits and the manuscript is much improved. Interesting study. The WGS data greatly improves the CIRM iPSC resource. Thank you.

Reviewer #2:

Remarks to the Author:

The authors have adequately addressed my comments. I am particularly pleased to see the extensive attention they now provide on the required sample-sizes to identify common variants that affect cell morphology. I believe this information is very valuable for the field, especially on future experimental designs.

Reviewer #3:

Remarks to the Author:

I am satisfied by the additional analyses and text changes, especially the sample size estimates.

NCOMMS-23-04923A Reviewer Rebuttal

Reviewers comments:

Reviewer #1 (Remarks to the Author):

The authors have done a thorough job with the edits and the manuscript is much improved. Interesting study. The WGS data greatly improves the CIRM iPSC resource. Thank you.

Reviewer #2 (Remarks to the Author):

The authors have adequately addressed my comments. I am particularly pleased to see the extensive attention they now provide on the required sample-sizes to identify common variants that affect cell morphology. I believe this information is very valuable for the field, especially on future experimental designs.

Reviewer #3 (Remarks to the Author):

I am satisfied by the additional analyses and text changes, especially the sample size estimates.

We thank the reviewers for their positive remarks on our updated manuscript.